# UHR-BAT: Budget-Aware Token Compression Vision-Language model for Ultra-High-Resolution Remote Sensing

Yunkai Dang[* 1]  Minxin Dai[* 1]  Yuekun Yang[1 2]  Zhangnan Li[3]  Wenbin Li[† 1]  Feng Miao[† 4]  Yang Gao[1]

## Abstract

Ultra-high-resolution (UHR) remote sensing imagery couples kilometer-scale context with query-critical evidence that may occupy only a few pixels. Such vast spatial scale leads to a quadratic explosion of visual tokens and hinders the extraction of information from small objects. Previous works utilize direct downsampling, dense tiling, or global top-K pruning, which either compromise query-critical image details or incur unpredictable compute. In this paper, we propose UHR-BAT, a query-guided and region-faithful token compression framework to efficiently select visual tokens under strict context budget. Specifically, we leverage text-guided, multi-scale importance estimation for visual tokens, effectively tackling the challenge of achieving precise yet low-cost feature extraction. Furthermore, by introducing region-wise preserve and merge strategies, we mitigate visual token redundancy, further driving down the computational budget. The experimental results show that UHR-BAT achieves state-of-the-art performance across various benchmarks. Code will be available at https://github.com/RL-MIND/UHR-BAT.

## 1. Introduction

Multimodal large language models (MLLMs) have rapidly advanced visual understanding by coupling robust image encoders with large language models (Yin et al., 2024; Liang et al., 2024; Wu et al., 2023; Kim et al., 2021; Yuan et al.,

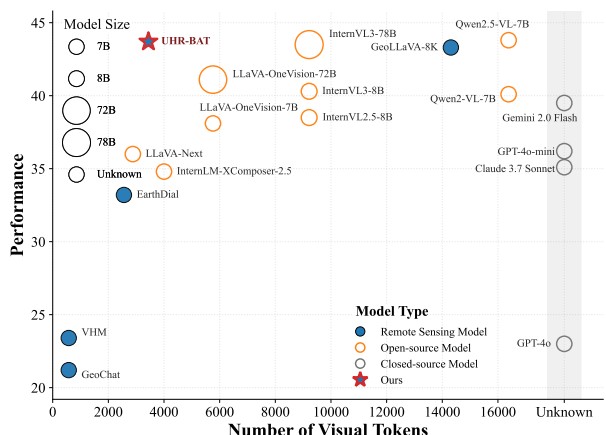

*Figure 1.* Accuracy–efficiency frontier for Ultra-High-Resolution Remote Sensing MLLMs on XLRS-Bench benchmark. We report overall weighted average accuracy versus the number of visual tokens. The results show that our method achieves superior performance using fewer tokens.

2021; Chen et al., 2022; Dang et al., 2024). In remote sensing, these models have demonstrated strong performance on geospatial question answering and reasoning tasks (Mall et al., 2023; Cong et al., 2022). However, real-world satellite and aerial imagery is typically ultra-high-resolution (UHR) (Liu et al., 2021; Yang et al., 2021). UHR visual question answering presents a critical challenge: images simultaneously encode kilometer-scale scene layouts, while the evidence required for reasoning is often extremely fine-grained (Li et al., 2024b). For instance, targets like small boats or vehicles may occupy only a few pixels within a multi-million-pixel image. Consequently, accurately localizing and reasoning about such small objects remains a critical challenge, yet it is essential for practical geospatial workflows.

In the domain of natural image processing, adaptive resolution and tiling-based strategies are widely adopted to extract details from high-resolution inputs (Guo et al., 2024; Zhang et al., 2024c; 2023). However, these pipelines often fragment global context and incur prohibitive computational costs as the number of tiles scales (Jaegle et al., 2021b;a). In the context of remote sensing, general MLLMs typically

---

[*]Equal contribution     † Corresponding author. [1]State Key Laboratory of Novel Software Technology, Nanjing University, Nanjing, China [2]Institute of Brain-inspired Intelligence, Nanjing University, Suzhou, China [3]School of Electronic Science and Engineering, Nanjing University [4]Institute of Brain-inspired Intelligence, National Laboratory of Solid State Microstructures, School of Physics, Nanjing University, Nanjing, China. Correspondence to: Wenbin Li, Feng Miao <liwenbin@nju.edu.cn, miao@nju.edu.cn>.

*Proceedings of the 43rd International Conference on Machine Learning*, Seoul, South Korea. PMLR 306, 2026. Copyright 2026 by the author(s).

resort to aggressive downsampling to address ultra-high-resolution (UHR) imagery. This compression sacrifices critical fine-grained details, thereby eliminating small-scale objects and obscuring semantic boundaries. Recent specialized frameworks attempt to address these limitations but face trade-offs (Wang et al., 2025a; Liu et al., 2025; Zhou et al., 2025; Luo et al., 2025). For instance, GeoLLaVA-8K (Wang et al., 2025a) employs visual pruning to discard background tokens. However, this text-agnostic approach frequently discards task-essential tokens while retaining substantial irrelevant information. Moreover, other works (Liu et al., 2025; Zhou et al., 2025; Luo et al., 2025) employ text-guided, coarse-to-fine zooming for token selection. However, this iterative regional selection process inevitably increases inference latency. Furthermore, it preserves a high volume of visually redundant tokens, limiting computational efficiency. Collectively, existing methods fail to process UHR imagery within a fixed token budget, as they lack mechanisms to prune redundancy while isolating query-relevant features.

To address these challenges, we propose a *Budget-Aware Token* Compression framework tailored for *UHR* Remote-Sensing MLLMs (UHR-BAT). UHR-BAT follows two principles. Compression should be *query-guided*, so that it retains evidence relevant to the question. Compression should also be *region-faithful*, so that it preserves coverage across semantically coherent regions and avoids suppressing sparse targets. Specifically, we introduce a Top-down query-guided mechanism that leverages text-aligned global priors to localize regions of interest. This ensures that fine-grained details are extracted exclusively from query-relevant areas, effectively avoiding visual redundancy. To further enforce region faithfulness, we propose the Region-wise Preserve and Merge (RPM) strategy to minimize redundancy by selectively preserving tokens in query-relevant areas while compressing the background. This strategy filters out irrelevant noise while maintaining essential evidence. As shown in Figure 1, our model achieves strong performance using substantially fewer visual tokens than other models. Experiments on the XLRS-Bench and RSHR-Bench benchmarks demonstrate that our model significantly enhances performance under constrained context budgets. Furthermore, this approach effectively shifts the accuracy-efficiency frontier for ultra-high-resolution remote sensing.

In summary, we highlight the primary novel contributions of this work as follows:

- We propose UHR-BAT, a token compression framework designed for ultra high resolution remote-sensing MLLMs under strict context budgets.

- We propose a query-guided, multi-scale input mechanism to integrate text-derived global priors and capture both holistic context and fine-grained details.

- We propose region-wise preserve and merge strategies to preserve salient local evidence and aggregate redundant background into compact representatives.

- Empirical results across standard benchmarks confirm that UHR-BAT establishes a new state-of-the-art for efficient UHR understanding, outperforming existing methods under strict token budgets.

## 2. Related Works

**Multimodal Large Language Models.** MLLMs integrate a vision encoder with an LLM through a projection or cross-attention interface, enabling unified multimodal understanding and generation (Alayrac et al., 2022; Ye et al., 2024; Dang et al., 2025a). Early models such as Flamingo (Alayrac et al., 2022) demonstrated few-shot multimodal prompting with tightly designed vision–language connectors. A complementary line aligns frozen components via lightweight bridging modules: BLIP-2 (Li et al., 2023) introduces the Q-Former to couple a frozen vision encoder with an LLM, and InstructBLIP (Dai et al., 2024) further improves instruction following with instruction-aware alignment. Building on these designs, open-source instruction-tuned models such as LLaVA (Li et al., 2024a) and MiniGPT-4 (Zhu et al., 2023) show that curated instruction data is critical for robust multimodal dialogue. Subsequent efforts such as ShareGPT4V (Chen et al., 2024a) scale and refine instruction data and training recipes, strengthening generalization and response quality. Recent models (Abdin et al., 2024; Hurst et al., 2024; Anthropic, 2025; Team et al., 2023; Li et al., 2024a) further enhance fine-grained and text-centric reasoning with stronger visual backbones and improved training strategies, including Qwen2.5-VL (Bai et al., 2025), InternVL3 (Zhu et al., 2025), and InternLM-XComposer (Zhang et al., 2023). In parallel, lightweight variants such as MobileVLM (Chu et al., 2023; 2024), TinyGPT-V (Yuan et al., 2023), LLaVA-Phi (Zhu et al., 2024), and MiniCPM-V (Yao et al., 2024) target efficient deployment under limited compute budgets.

**Remote Sensing Foundation Models.** Remote-sensing vision-language models apply open-vocabulary reasoning to earth observation. RSGPT (Hu et al., 2025) pioneers RS-specific conversational modeling for captioning and QA. GeoChat (Kuckreja et al., 2024) enhances grounding by enabling region-specific prompts and outputting bounding boxes. EarthGPT (Zhang et al., 2024b) and EarthDial (Soni et al., 2025) unify multiple interpretation tasks within a single interface. Similarly, SkyEyeGPT (Zhan et al., 2025) extends instruction tuning for broad task coverage, while RS-LLaVA (Bazi et al., 2024) adapts general VLM frameworks for RS captioning and VQA. To address reliability, H2RSVLM (Pang et al., 2024) explicitly models honesty

and rejects unanswerable queries. Other works incorporate structured knowledge; SkySenseGPT (Luo et al., 2024b) utilizes scene-graph supervision, and LHRS-Bot (Muhtar et al., 2024) leverages VGI-enhanced signals for alignment. MF-RSVLM (Dang et al., 2025b) addresses the loss of fine-grained details by integrating multi-scale feature fusion and employing a recurrent mechanism to reinject visual signals during long-context generation. However, these models are limited to low-resolution datasets, typically processing images with resolution smaller than $1024 \times 1024$ pixels.

**High-Resolution MLLMs in General Domains.** High-resolution MLLMs are fundamentally constrained by token explosion and the resulting attention and memory overhead when preserving fine-grained structures (Yang et al., 2026). In general domains, LLaVA-UHD (Guo et al., 2024) processes arbitrary aspect ratios via resolution-aware slicing and token-efficient encoding, and LLaVA-UHD v2 (Zhang et al., 2024c) further improves multi-scale perception with feature pyramids and hierarchical window attention. However, pipelines based on slicing or tiling often fragment global context and incur increased computation as the number of tiles grows. To alleviate cross-slice discontinuity, HiRes-LLaVA (Huang et al., 2025) introduces restoration and aggregation mechanisms, and InternLM-XComposer2-4KHD (Dong et al., 2024) supports dynamic resolutions up to 4K via flexible patch configurations for text-centric understanding. Nevertheless, these methods are generally not query-adaptive and still spend substantial tokens and compute on redundant regions. In remote sensing, GeoLLaVA-8K (Wang et al., 2025a) scales to $8K \times 8K$ by exploiting background redundancy through background token pruning and anchored token selection, while ZoomEarth (Liu et al., 2025) and ZoomSearch (Zhou et al., 2025) adopt coarse-to-fine zooming to recover answer-critical details. In contrast, our method enforces a strict visual-token budget. It preserves both global context and local evidence within a fixed-length sequence.

# 3. Method

## 3.1. Overview

High-resolution remote sensing imagery demands simultaneously modeling (i) *global context* such as large-scale land-use layout and long-range spatial dependencies, and (ii) *fine-grained evidence* such as small objects and sharp boundaries. Directly feeding high-resolution visual tokens into Multimodal Large Language Models (MLLMs) is infeasible: standard vision encoders produce prohibitively long token sequences that exceed the limited context length of modern MLLMs. Simple remedies (e.g., heavy downsampling, tiling, or global top-$K$ pruning) either compromise global coherence or suppress localized cues. We therefore seek a *query-guided* compression strategy that removes redundancy while remaining *region-faithful* to small targets and structurally diverse scenes.

Our method comprises three components, as illustrated in Figure 2. First, we compute query-aware visual token importance from the vision–language attention interface (§3.2) and use it as a common scoring signal across views. Second, we construct multi-scale token streams with *scale-specific positional embedding* and *cross-scale importance alignment* to make scores comparable across resolutions (§3.3). Third, we introduce Region-wise Preserve-and-Merge, which preserves informative tokens within each region while merging redundant ones into compact representatives (§3.4).

## 3.2. Preliminaries

**MLLM Architecture.** We consider a generic MLLM parameterized by $\theta$, consisting of a vision encoder $\mathcal{V}$, a text embedding layer $\mathcal{T}$, a vision–language projector $\phi$, and a stack of $L$ transformer layers. Given a high-resolution image $I \in \mathbb{R}^{H \times W \times 3}$ and a textual query $\mathbf{x}$, the vision encoder outputs a grid of visual features $F = \mathcal{V}(I) \in \mathbb{R}^{N \times d_v}$, where $N = U \cdot V$ is the number of visual tokens on a $U \times V$ grid. We map them into the model embedding space via $\phi : \mathbb{R}^{d_v} \to \mathbb{R}^d$, yielding visual tokens

$$E = \{\mathbf{e}_i\}_{i=1}^N = \phi(F), \qquad \mathbf{e}_i \in \mathbb{R}^d. \qquad (1)$$

Each token $i$ is also associated with a 2D coordinate $\mathbf{pos}_i \in [0, 1]^2$ given by the center of its receptive field on the token grid. In parallel, the query is embedded into text tokens $T = \mathcal{T}(\mathbf{x}) \in \mathbb{R}^{M \times d}$. The transformer processes a concatenated sequence of visual tokens $E$ and text tokens $T$, denoted as $\text{Concat}(E, T)$. The model then generates the response in an autoregressive manner.

**Query-aware Token Importance.** We quantify the relevance of each visual token $\mathbf{e}_i$ to the query using text-to-vision attention weights. Let $\mathbf{A} \in \mathbb{R}^{M \times N}$ denote the cross-modal attention matrix extracted from the vision–language attention interface (we average attention over heads and use a fixed interface layer in our implementation). We aggregate attention across text tokens to obtain a scalar importance score for each visual token:

$$a_i = \frac{1}{M} \sum_{j=1}^M \mathbf{A}_{j,i}. \qquad (2)$$

**Token Budget.** Due to the limited context length, we impose a hard cap $B \ll N$ on the number of *visual* tokens passed to the language model (text tokens and generated tokens are budgeted separately in practice). To address this, we define a compression operator $C(\cdot)$ that maps dense visual tokens into a budget-feasible sequence:

$$\bar{E} = C(E; B, \{a_i\}_{i=1}^N), \qquad \text{s.t.} \quad |\bar{E}| \leq B. \qquad (3)$$

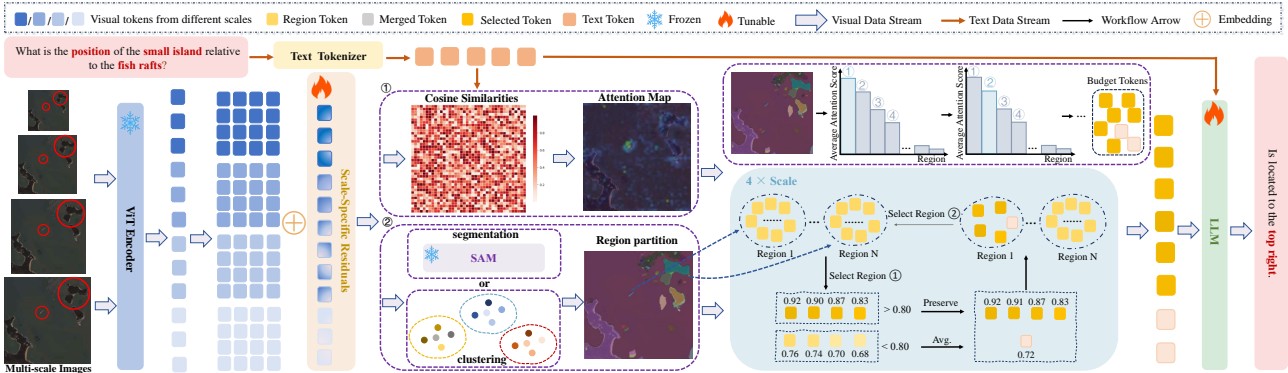

*Figure 2.* **Overview of our method.** We encode a high-resolution remote sensing image at multiple scales using a frozen ViT, applying Scale-Specific Positional Embeddings to distinguish visual tokens across scales, and then obtain an anchor-scale query-to-vision attention map from the MLLM interface. A region partition (e.g., induced by SAM or feature+coordinate clustering) enables region-wise preserve-and-merge, while remaining tokens are merged via average pooling to retain coarse context. A final top-$k$ step enforces the per-scale token budget. Finally, the processed sequence is fed into the LLM to generate the final answer.

In §3.4, we instantiate $C(\cdot)$ via Region-wise Preserve-and-Merge. We further introduce an explicit budget enforcement step to ensure that the constraints are strictly satisfied.

**Region Partition.** Pure importance-based pruning often retains tokens encoding overlapping information. These redundant tokens consume a significant fraction of the budget. Under a hard cap, this biases the selection toward dominant regions, causing sparse yet critical targets to be missed. To encourage structural diversity and preserve fine-grained details, we partition the token index set $\mathcal{I} = \{1, \ldots, N\}$ at the original resolution into $R$ disjoint regions:

$$\mathcal{P} = \{\mathcal{S}_m\}_{m=1}^R, \qquad \bigcup_{m=1}^R \mathcal{S}_m = \mathcal{I}, \quad \mathcal{S}_m \cap \mathcal{S}_n = \emptyset. \quad (4)$$

Our goal is to group tokens that convey the same semantic information. Therefore, we instantiate the partition using methods that detect such similarities. Options include clustering in a joint feature-coordinate space (McQueen, 1967; **?**; Lloyd, 1982), or using segmentation models (e.g., SAM (Kirillov et al., 2023)) followed by a token-level mapping (Appendix C.2). We apply this independently at each scale.

### 3.3. Scale-Specific Positional Embedding

To capture both global context and fine details, we construct $S$ resized views $\{I^{(s)}\}_{s=1}^S$. We designate the lowest-resolution view ($s = 1$) as the *anchor* to preserve global structure. Higher scales provide progressively finer details. For each scale $s$, the vision encoder outputs tokens $E^{(s)} = \{\mathbf{e}_i^{(s)}\}_{i=1}^{N_s}$. These tokens form a $U_s \times V_s$ grid, where $N_s = U_s \cdot V_s$. Tokens from different resolutions must remain spatially aligned and distinguishable. Therefore, we

augment each token to obtain updated features:

$$\mathbf{h}_i^{(s)} = \mathbf{e}_i^{(s)} + \mathbf{p}_i^{(s)} + \mathbf{q}^{(s)}. \quad (5)$$

Here $\mathbf{p}_i^{(s)} \in \mathbb{R}^d$ is obtained by bilinearly interpolating a base 2D positional embedding (defined on the pretraining grid) to the $U_s \times V_s$ grid, and $\mathbf{q}^{(s)} \in \mathbb{R}^d$ is a learned scale embedding (a lookup table over $s$). Interpolation preserves geometric consistency across resized views, and $\mathbf{q}^{(s)}$ prevents ambiguity between tokens from different scales.

**Cross-scale Importance Alignment.** Token selection should be comparable across scales under a unified notion of importance. We take the anchor-scale importance as reference: we reshape the anchor token scores $\{a_i^{(1)}\}$ onto the $U_1 \times V_1$ grid to form $\mathbf{A}^{(1)} \in \mathbb{R}^{U_1 \times V_1}$. For any other scale $s$, let $u \in \{1, \ldots, U_s\}$ and $v \in \{1, \ldots, V_s\}$ denote a grid location on the $U_s \times V_s$ token grid. We map this location to a continuous coordinate on the anchor grid $U_1 \times V_1$ using the deterministic cell-center resize mapping

$$\tau_s(u, v) = \left( \frac{u - \frac{1}{2}}{U_s} U_1 + \frac{1}{2}, \frac{v - \frac{1}{2}}{V_s} V_1 + \frac{1}{2} \right), \quad (6)$$

When $s = 1$, this mapping reduces to the identity at the anchor scale. We then define

$$\mathbf{A}_{u,v}^{(s)} = \Psi\left( \mathbf{A}^{(1)}, \tau_s(u, v) \right), \quad (7)$$

where $\Psi(\cdot)$ denotes bilinear interpolation (Appendix C.3). Each token $i$ at scale $s$ covers a set of grid cells $\Omega_i^{(s)}$ (typically one cell for patch tokens), and we compute its aligned importance score by averaging:

$$a_i^{(s)} = \frac{1}{|\Omega_i^{(s)}|} \sum_{(u,v) \in \Omega_i^{(s)}} \mathbf{A}_{u,v}^{(s)}. \quad (8)$$

Fixing importance to one reference map and transferring it across scales yields comparable scores $\{a_i^{(s)}\}$, which is crucial for per-scale budgeting.

**Per-scale Budgets.** Given a global limit $B$, we allocate a budget $\{B_s\}$ to each scale $s$ such that $\sum_{s=1}^{S} B_s \leq B$. Higher resolutions correspond to larger $N_s$ and provide more detailed information. Therefore, we assign larger $B_s$ values to these scales. We also enforce a minimum capacity $B_s \geq R_s$. As the region partition $\mathcal{P}^{(s)}$ utilizes high-resolution features independent of the anchor attention, this constraint effectively compensates for potential deviations in attention scores, mathematically guaranteeing that all selected semantic regions retain at least one merged representative token. Finally, we compress each scale into $\bar{E}^{(s)} = C(E^{(s)}; B_s, \{a_i^{(s)}\})$ with $|\bar{E}^{(s)}| \leq B_s$.

### 3.4. Region-wise Preserve-and-Merge

Small targets in high-resolution remote sensing imagery are intrinsically sparse. Global top-$K$ strategies tend to concentrate the budget on visually dominant regions. They often retain excessive tokens representing repetitive information. Consequently, critical local evidence is marginalized, especially under low budget conditions. We address this by enforcing a *region-wise* retention rule. Within each region, we preserve relatively important tokens. The remaining redundant tokens are merged into a compact representative.

We operate on each scale $s$ independently. Consider the region partition $\mathcal{P}^{(s)} = \{\mathcal{S}_m^{(s)}\}_{m=1}^{R_s}$. For each region $m$, we calculate its average importance $\mu_m^{(s)}$:

$$\mu_m^{(s)} = \frac{1}{|\mathcal{S}_m^{(s)}|} \sum_{i \in \mathcal{S}_m^{(s)}} a_i^{(s)}. \qquad (9)$$

Subsequently, we retain tokens with importance no lower than this region-specific reference:

$$\mathcal{K}_m^{(s)} = \left\{ i \in \mathcal{S}_m^{(s)} \mid a_i^{(s)} \geq \mu_m^{(s)} \right\}, \qquad (10)$$

and define the redundancy set as $\mathcal{R}_m^{(s)} = \mathcal{S}_m^{(s)} \setminus \mathcal{K}_m^{(s)}$. By construction, $\mathcal{K}_m^{(s)}$ is non-empty for any region with real-valued scores (Appendix C.4). This guarantees that every region contributes at least one informative token.

Rather than discarding $\mathcal{R}_m^{(s)}$, we merge it into a single representative token via mean pooling:

$$\tilde{\mathbf{e}}_m^{(s)} = \frac{1}{|\mathcal{R}_m^{(s)}|} \sum_{i \in \mathcal{R}_m^{(s)}} \mathbf{h}_i^{(s)}, \qquad \text{if } |\mathcal{R}_m^{(s)}| > 0. \qquad (11)$$

**Priority-based Serialization and Budgeting.** To maximize information density, we enforce a serialization order that prioritizes regions with higher average importance. First,

we sort the regions in descending order of their average scores $\mu_m^{(s)}$. Let $\pi(k)$ denote the index of the region with the $k$-th highest average importance.

Within each region, we arrange the retained tokens by descending importance $a_i^{(s)}$ to form a sorted local sequence:

$$\mathbf{S}_{\pi(k)}^{(s)} = \{\mathbf{h}_i^{(s)} : i \in \mathcal{K}_{\pi(k)}^{(s)}\}_\downarrow, \qquad (12)$$

where $\{\cdot\}_\downarrow$ indicates sorting in descending order based on internal token importance. We then construct the global candidate sequence by concatenating these ordered regional groups alongside their merged tokens:

$$E_{\text{cand}}^{(s)} = \text{Concat}_{k=1}^{R_s} \left( \mathbf{S}_{\pi(k)}^{(s)}, \{\tilde{\mathbf{e}}_{\pi(k)}^{(s)} : |\mathcal{R}_{\pi(k)}^{(s)}| > 0\} \right), \qquad (13)$$

where $\tilde{\mathbf{e}}_{\pi(k)}^{(s)}$ is appended to the end of each regional group. Finally, we enforce the budget $B_s$ by selecting the top $B_s$ tokens from $E_{\text{cand}}^{(s)}$. This ensures that the most critical regions and tokens are prioritized within the hard capacity limit.

### 3.5. Overall

The overall pipeline of our region-aware token pruning approach is summarized in Appendix C.1. Given the input visual tokens $\mathbf{X} \in \mathbb{R}^{N \times C}$, we first partition them into distinct semantic regions $\mathcal{P} = \{\mathcal{S}_1, \ldots, \mathcal{S}_R\}$ using either the K-means-based or SAM-induced method described in Appendix C.2. Within each region $\mathcal{S}_r$, we evaluate token importance and retain only the most informative tokens set $\mathcal{K}_r$, effectively pruning the redundant background tokens. The final output is the concatenation of retained tokens from selected regions, preserving the spatial structure and semantic integrity with reduced computational cost.

## 4. Experiment

**Datasets and Models.** We evaluate on three complementary high-resolution remote-sensing benchmarks: MME-RealWorld-RS (Fu et al., 2025), RSHR-Bench (Dang et al., 2025c), and XLRS-Bench (Wang et al., 2025b). MME-RealWorld-RS measures multiple-choice VQA skills on real-world overhead imagery, including position understanding, color recognition, and counting, and we report accuracy for each skill as well as the averaged score (Table 3). RSHR-Bench is a super-high-resolution benchmark built on native 4K+ remote-sensing imagery. In this work, we follow its multiple-choice VQA protocol and report category-wise accuracies over Perception and Reasoning tasks together with their averages. XLRS-Bench serves as our primary standardized suite for extremely large remote-sensing images, where we follow the official aggregation to report overall performance alongside representative sub-tasks. We consider three model families, including open-source general-purpose MLLMs, closed-source MLLMs,

and remote-sensing-specific VLMs. All models are evaluated in a zero-shot setting with a unified prompt template. For completeness, the task definitions, evaluation protocols, and table column abbreviations are provided in Appendix B.

**Main Results on XLRS-Bench.** To verify the effectiveness of our method, we evaluate our method on XLRS-Bench (Wang et al., 2025b) across the perception and reasoning dimensions. As shown in Table 1, we follow the official protocol with a uniform prompt for all sub-tasks and report the overall accuracy weighted by the benchmark's sub-task sample counts (w.Avg.). Our model achieves a state-of-the-art performance of 44.0 w.Avg. across open-source and closed-source models. Relative to prior remote-sensing MLLMs, our method improves the overall weighted accuracy by +22.8% over GeoChat (Kuckreja et al., 2024), +0.7% over GeoLLaVA-8K (Wang et al., 2025a), +10.2% over EarthDial (Soni et al., 2025), and +20.4% over VHM (Pang et al., 2025). We also outperform strong closed-source models, yielding +21.0% and +8.9% gains over GPT-4o (Hurst et al., 2024) and Claude 3.7 Sonnet (Anthropic, 2025), respectively. Moreover, we surpass recent large open-source vision-language models, with improvements of +2.9% over LLaVA-OneVision-72B (Li et al., 2024a). Overall, these gains across both perception and reasoning tasks indicate that our method enhances multimodal capability for high-resolution remote sensing scenarios.

**Main Results on RSHR-Bench.** To verify higher-resolution remote sensing understanding, we further evaluate our method on the RSHR-Bench dataset (Dang et al., 2025c), which measures performance along three facets: *Perception*, *Reasoning*, and *Multi-turn* interaction. As shown in Table 2, our method achieves the best Perception average among open-source and remote-sensing baselines, with 29.2 on Perception, and obtains a strong Reasoning average of 45.0, outperforming all remote-sensing VLMs. On Perception, our average improves over EarthDial (Soni et al., 2025), GeoChat (Kuckreja et al., 2024), GeoLLaVA-8K (Wang et al., 2025a), and VHM (Pang et al., 2025) by +1.1%, +3.3%, +4.7%, and +3.5%, respectively, and also exceeds the representative 4K image input open-source baseline VILA-HD (Shi et al., 2025) by +1.2%. On Reasoning, our average yields clear gains over remote-sensing VLMs, improving by +7.9% over EarthDial, +16.7% over GeoChat, +15.0% over GeoLLaVA-8K, and +17.3% over VHM, while remaining competitive with strong general VLMs. Beyond aggregate scores, our model achieves state-of-the-art results across several key metrics. Specifically, we obtain the best performance on MRJC (65.0), SHP (32.0), and DET (38.0), outperforming the next strongest baselines by margins of +10.0%, +4.0%, and +8.0%, respectively. These consistent improvements underscore our model's superior region-aware detection and reasoning capabilities. Overall, these results demonstrate improved robustness for high-resolution

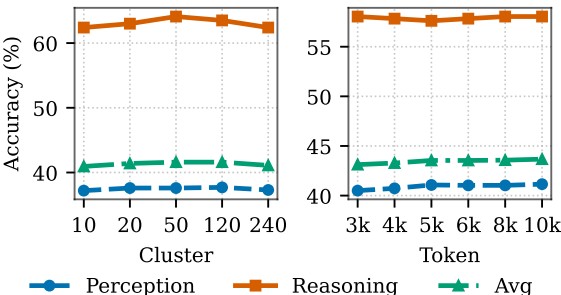

*Figure 3.* Ablation on clustering centers and retained tokens. **Left:** accuracy versus the number of cluster centers $k$. **Right:** accuracy versus the number of retained visual tokens.

remote sensing perception and understanding, particularly under iterative, region-conditioned inference.

**Main Results on MMERealworld Benchmark.** We further evaluate our method on widely-used benchmarks in this work, and report results on the MMERealworld-RS dataset. As shown in Table 3, we evaluate our method by the mean accuracy over three perception-oriented sub-tasks (Position, Color, and Count). Overall, we observe that scaling the vision backbone and increasing input resolution consistently enhance spatial localization and color discrimination. Our method achieves the highest average score (Avg. = 33.33), demonstrating robust performance in position (44.00) and color (42.00) tasks. In terms of Avg., our method improves over remote-sensing VLMs by +12.01% relative to GeoChat, +9.15% relative to VHM, and +4.92% relative to GeoLLaVA-8K, and it also exceeds the closed-source GPT-4o baseline by +5.91%. The gains are particularly pronounced on the perceptual dimensions: compared with the strongest remote-sensing baseline, we improve position by +8.76% and color by +14.08%.

**Ablation on Cluster Centers and Retained Tokens.** We ablate two key hyper-parameters in our pipeline: (i) the number of cluster centers $k$ used in our clustering-based method (Figure 3, left), and (ii) the number of visual tokens retained (Figure 3, right). We report accuracy on Perception, Reasoning, and the overall Avg score on XLRS-Bench (Wang et al., 2025b). As shown in the left plot, the performance displays a gentle rise-and-fall pattern. In particular, Reasoning exhibits a mild peak at a moderate $k$, while perception remains nearly unchanged across the tested range. As illustrated in the right plot, increasing the token budget consistently yields performance gains, validating the effectiveness of our token selection strategy. However, the performance curves for both Perception and Reasoning rapidly saturate. This stability suggests that our model is highly efficient at extracting the most semantic-rich visual tokens, achieving near-optimal performance even under constrained budgets. Overall, these results demonstrate that our method is capable of capturing the vast majority of essential visual features.

*Table 1.* **Results on XLRS-Bench across perception and reasoning sub-tasks.** 'w.Avg.' indicates the sample-count-weighted average. The best results are **bolded**, and the second-best results are underlined. Full sub-task names are provided in the appendix.

| Method | Size | Perception | | | | | | | | Reasoning | | | | | Overall |
|---|---|---|---|---|---|---|---|---|---|---|---|---|---|---|---|
| Sub-tasks (L-3 Capability) | | OC | RC | OLUC | RLUC | OCC | OCL | OMS | OSR | AD | ECR | RP | RCCD | CCR | w.Avg. |
| number of samples | | 60 | 100 | 100 | 200 | 800 | 800 | 60 | 500 | 100 | 100 | 100 | 60 | 100 | 3080 |
| *Open-source MLLMs* | | | | | | | | | | | | | | | |
| InternLM-XComposer-2.5 (Zhang et al., 2024a) | 7B | 21.7 | 42.0 | 7.0 | 68.0 | 31.8 | 27.8 | 6.7 | 26.0 | 72.0 | 81.0 | 41.0 | 36.7 | 47.0 | 34.8 |
| LLaVA-Next (Li et al., 2024a) | 7B | 26.7 | 40.0 | 5.0 | 67.0 | 28.8 | 32.8 | 66.7 | 30.0 | 69.0 | 78.0 | 27.0 | 35.0 | 36.0 | 36.0 |
| LLaVA-OneVision-7B (Li et al., 2024a) | 7B | 25.0 | 38.0 | 2.0 | 69.5 | 35.9 | 35.3 | 65.0 | 25.2 | 76.0 | 83.0 | 24.0 | 43.3 | 36.0 | 38.1 |
| InternVL2.5-8B (Chen et al., 2024b) | 8B | 38.3 | 37.0 | 10.0 | 77.0 | 33.4 | 35.5 | 65.0 | 21.6 | 73.0 | 83.0 | 34.0 | 50.0 | 43.0 | 38.5 |
| InternVL3-8B (Chen et al., 2024c) | 8B | 40.0 | 39.0 | 10.0 | 71.5 | 44.5 | 30.8 | 65.0 | 25.2 | 77.0 | 82.0 | 36.0 | 21.7 | 50.0 | 40.3 |
| Qwen2-VL-7B (Wang et al., 2024) | 7B | 26.7 | 40.0 | 11.0 | 73.0 | 35.9 | 34.6 | 61.7 | 31.8 | 70.0 | 81.0 | 35.0 | 46.7 | 48.0 | 40.1 |
| Qwen2.5-VL-7B (Bai et al., 2025) | 7B | 33.3 | 40.0 | 31.0 | 77.0 | 40.6 | 40.5 | 66.7 | 36.2 | 68.0 | 72.0 | 27.0 | 38.3 | 45.0 | 43.8 |
| LLaVA-OneVision-72B (Li et al., 2024a) | 72B | 33.3 | 38.0 | 15.0 | 72.5 | 36.3 | 36.3 | 66.7 | 35.6 | 74.0 | 83.0 | 28.0 | 36.7 | 43.0 | 41.1 |
| InternVL3-78B (Zhu et al., 2025) | 78B | 23.3 | 49.0 | 33.0 | 74.0 | 42.5 | 37.4 | 66.7 | 30.0 | 76.0 | 81.0 | 40.0 | 45.0 | 42.0 | 43.5 |
| *Closed-source MLLMs* | | | | | | | | | | | | | | | |
| GPT-4o (Hurst et al., 2024) | – | 25.0 | 32.0 | 15.0 | 66.0 | 9.5 | 11.3 | 11.7 | 24.6 | 73.0 | 73.0 | 35.0 | 20.0 | 25.0 | 23.0 |
| GPT-4o-mini (Hurst et al., 2024) | – | 23.3 | 25.0 | 19.0 | 59.5 | 40.9 | 31.0 | 65.0 | 23.6 | 71.0 | 71.0 | 29.0 | 6.7 | 30.0 | 36.2 |
| Claude 3.7 Sonnet (Anthropic, 2025) | – | 27.6 | 22.7 | 17.4 | 68.4 | 30.5 | 29.9 | 63.6 | 27.6 | 64.8 | 78.4 | 34.5 | 27.8 | 32.6 | 35.1 |
| Gemini 2.0 Flash (Team et al., 2023) | – | 41.7 | 45.0 | 38.0 | 73.5 | 34.6 | 27.6 | 61.7 | 32.0 | 73.0 | 82.0 | 43.0 | 30.0 | 51.0 | 39.5 |
| *Remote Sensing MLLMs* | | | | | | | | | | | | | | | |
| GeoChat (Kuckreja et al., 2024) | 7B | 16.7 | 29.0 | 2.0 | 23.0 | 21.1 | 16.8 | 35.0 | 24.2 | 33.0 | 43.0 | 10.0 | 24.0 | 21.0 | 21.2 |
| GeoLLaVA-8K (Wang et al., 2025a) | 7B | 26.7 | 38.0 | 49.0 | 69.0 | 41.6 | 31.6 | 65.0 | 35.0 | 67.0 | 78.0 | 66.0 | 50.0 | 52.0 | 43.3 |
| EarthDial (Soni et al., 2025) | 7B | 18.3 | 42.0 | 1.0 | 36.0 | 31.3 | 31.0 | 65.0 | 24.8 | 62.0 | 71.0 | 43.0 | 48.3 | 50.0 | 33.8 |
| VHM (Pang et al., 2025) | 7B | 16.7 | 30.0 | 2.0 | 26.0 | 21.4 | 16.8 | 35.0 | 25.6 | 42.0 | 53.0 | 46.0 | 28.3 | 21.0 | 23.6 |
| **UHR-BAT (Ours)** | 7B | 21.7 | 33.0 | **50.0** | 55.5 | 43.5 | 33.8 | 65.0 | **44.8** | 62.0 | 71.0 | 54.0 | 46.7 | 51.0 | **44.0** |

*Table 2.* **RSHR-Bench results across Perception and Reasoning tasks.** We report accuracy (%) for each subtask. 'P.Avg.' and 'R.Avg.' denote the mean accuracy over subtasks within Perception and Reasoning, respectively. 'All Avg.' denotes the mean accuracy over all Perception and Reasoning subtasks. Subtask abbreviations are defined in the appendix.

| Model | Size | Perception | | | | | | | | | Reasoning | | | | | Average | | |
|---|---|---|---|---|---|---|---|---|---|---|---|---|---|---|---|---|---|---|
| | | COL | SHP | DET | OC | REL | OGD | RG | OCN | RCN | AD | FP | MRJC | MRJCS | OSJ | P.Avg. | R.Avg. | Avg. |
| *Open-source VLMs* | | | | | | | | | | | | | | | | | | |
| InternVL2.5-8B (Chen et al., 2024b) | 8B | 25.5 | 22.0 | 26.0 | 26.0 | 22.5 | 24.5 | 30.0 | 22.5 | 20.0 | 26.0 | 20.0 | 22.5 | 34.0 | 20.0 | 24.3 | 24.5 | 24.4 |
| InternVL3.5-8B (Wang et al., 2025c) | 8B | 21.5 | 28.0 | 18.0 | 21.5 | 29.0 | 28.5 | 30.0 | 26.5 | 25.0 | 20.0 | 16.0 | 29.0 | 34.0 | 26.0 | 25.3 | 25.0 | 25.2 |
| MiniCPM2_6 (Yao et al., 2024) | 7B | 21.5 | 28.0 | 30.0 | 24.0 | 19.5 | 29.5 | 34.3 | 22.0 | 29.0 | 26.0 | 30.0 | 35.0 | 32.0 | 30.0 | 27.4 | 30.6 | 28.5 |
| Phi-3.5-Vision (Abdin et al., 2024) | 7B | 25.0 | 24.0 | 25.0 | 25.0 | 23.5 | 25.0 | 22.9 | 25.0 | 25.0 | 24.0 | 22.0 | 23.5 | 30.0 | 22.0 | 24.5 | 24.3 | 24.4 |
| Qwen2.5-VL-7B (Bai et al., 2025) | 7B | 29.5 | 25.0 | 22.0 | 28.0 | 25.0 | 24.5 | 24.3 | 26.5 | 22.0 | 26.0 | 28.0 | 25.0 | 10.0 | 20.0 | 25.2 | 21.8 | 24.0 |
| DeepSeek-VL (Lu et al., 2024) | 7B | 22.5 | 22.0 | 21.0 | 25.0 | 20.5 | 26.0 | 28.6 | 20.5 | 22.0 | 22.0 | 28.0 | 50.0 | 32.0 | 20.0 | 23.1 | 30.4 | 25.7 |
| VILA-HD (Shi et al., 2025) | 7B | 40.0 | 22.0 | 22.0 | 37.0 | 35.5 | 26.0 | 21.4 | 24.5 | 24.0 | 58.0 | 30.0 | 55.0 | 32.0 | 58.0 | 28.0 | 46.6 | 34.6 |
| *Closed-source VLMs* | | | | | | | | | | | | | | | | | | |
| GPT-5 (OpenAI, 2025) | – | 29.0 | 10.0 | 23.0 | 23.0 | 37.0 | 24.5 | 31.4 | 20.0 | 23.0 | 74.0 | 58.0 | 35.0 | 34.0 | 66.0 | 24.5 | 53.4 | 34.8 |
| GPT-4o (Hurst et al., 2024) | – | 49.5 | 23.0 | 15.0 | 35.5 | 30.5 | 28.0 | 27.1 | 22.5 | 41.0 | 68.0 | 56.0 | 30.5 | 32.0 | 64.0 | 30.2 | 50.1 | 37.3 |
| GPT-4o-mini (Hurst et al., 2024) | – | 41.5 | 16.0 | 29.0 | 31.5 | 31.5 | 32.0 | 28.6 | 19.5 | 32.0 | 54.0 | 54.0 | 31.5 | 48.0 | 54.0 | 29.1 | 48.3 | 36.0 |
| *Remote Sensing VLMs* | | | | | | | | | | | | | | | | | | |
| EarthDial (Soni et al., 2025) | 7B | 41.0 | 22.0 | 21.0 | 30.0 | 32.5 | 30.5 | 27.1 | 18.0 | 31.0 | 42.0 | 30.0 | 29.5 | 32.0 | 52.0 | 28.1 | 37.1 | 31.3 |
| GeoChat (Kuckreja et al., 2024) | 7B | 32.5 | 22.0 | 24.0 | 29.5 | 40.0 | 25.0 | 22.9 | 22.5 | 29.0 | 30.0 | 24.0 | 25.5 | 30.0 | 32.0 | 25.9 | 28.3 | 26.8 |
| GeoLLaVA-8K (Wang et al., 2025a) | 7B | 25.0 | 25.0 | 25.0 | 25.0 | 25.0 | 25.0 | 21.4 | 25.0 | 25.0 | 24.0 | 30.0 | 40.0 | 34.0 | 20.0 | 24.5 | 30.0 | 26.5 |
| VHM (Pang et al., 2025) | 7B | 25.5 | 25.0 | 26.0 | 26.5 | 55.0 | 25.0 | 22.9 | 25.0 | 25.0 | 26.0 | 24.0 | 26.5 | 34.0 | 28.0 | 25.7 | 27.7 | 26.4 |
| **UHR-BAT (Ours)** | 7B | **46.5** | **32.0** | **38.0** | 12.5 | 38.5 | 24.0 | 17.1 | 17.0 | 37.0 | 44.0 | 34.0 | **65.0** | 32.0 | 50.0 | **29.2** | 45.0 | **34.8** |

*Table 3.* **Results on the MMERealworld-RS dataset, with methods ranked by Avg.** We report accuracy (%) on Position, Color, and Count, and their mean (Avg.) for all methods. 'Params' denotes the number of model parameters.

| Method | Params | Position | Color | Count | Avg. |
|---|---|---|---|---|---|
| GPT-4o (Hurst et al., 2024) | – | 33.52 | 29.83 | 18.90 | 27.42 |
| LLaVa-v1.5-7B (Liu et al., 2023) | 7B | 21.48 | 22.95 | 16.31 | 20.28 |
| LLaVA-v1.6-7B (Li et al., 2024a) | 7B | 26.49 | 24.06 | 20.47 | 23.70 |
| LLaVA-ov-7B (Li et al., 2024a) | 7B | 26.81 | 26.14 | 27.57 | 26.83 |
| Qwen2.5-VL-7B (Bai et al., 2025) | 7B | 22.12 | 15.54 | 14.93 | 17.55 |
| LLaVA-HR (Luo et al., 2024a) | 7B | 35.56 | 44.30 | 7.91 | 29.26 |
| GeoChat (Kuckreja et al., 2024) | 7B | 25.06 | 23.11 | 15.66 | 21.32 |
| VHM (Pang et al., 2025) | 7B | 35.24 | 20.32 | 16.80 | 24.18 |
| GeoLLaVA-8K (Wang et al., 2025a) | 7B | 34.90 | 27.92 | 22.27 | 28.41 |
| **UHR-BAT (Ours)** | 7B | **44.00** | 42.00 | 14.00 | **33.33** |

**Computational Efficiency and Inference Latency.** Table 4 provides a granular analysis of the computational overhead under varying token budgets. Given the massive input size typical of high-resolution remote sensing imagery (totaling 131,328 tokens in this baseline), processing the full sequence is often computationally prohibitive. Our dynamic token compression mechanism demonstrates significant efficiency gains: by increasing the compression ratio from $10.94\times$ (12k tokens) to $32.83\times$ (4k tokens), we observe a linear reduction in theoretical computational cost (TFLOPs drop from 374.24 to 267.68). Crucially, this reduction translates into tangible improvements in inference throughput, with generation latency decreasing by approximately **27.5%** (from 5.99s to 4.34s). These results confirm that our method can effectively decouple input resolution from computational cost, enabling the deployment of our model on resource-constrained platforms.

**Scaling Behavior of Token Selection at Multiple Resolutions.** Table 6 reports the performance on XLRS-

*Table 4.* Efficiency analysis: Latency and FLOPs per sample under different token budgets (Total input tokens = 131,328).

| Compression ($\times$) | Kept Tokens | TFLOPs | Avg. Total (s) | Avg. Gen. (s) |
|---|---|---|---|---|
| 32.83 | 4k | 267.68 | 20.93 | 4.34 |
| 26.27 | 5k | 274.91 | 21.58 | 4.56 |
| 21.89 | 6k | 289.10 | 21.83 | 4.74 |
| 16.42 | 8k | 317.48 | 21.99 | 5.11 |
| 13.13 | 10k | 345.75 | 25.50 | 5.65 |
| 11.94 | 11k | 359.94 | 25.79 | 5.86 |
| 10.94 | 12k | 374.24 | 22.79 | 5.99 |

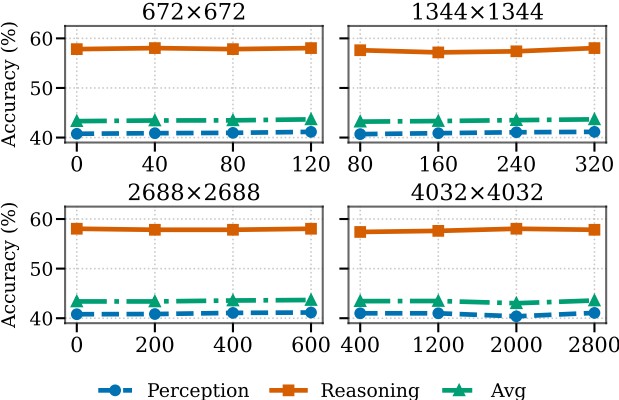

*Figure 4.* Multi-scale token selection accuracy on XLRS-Bench. We evaluate four input resolutions (672×672, 1344×1344, 2688×2688, 4032×4032) and vary the number of retained visual tokens at each scale (x-axis).

Bench across four resolutions, ranging from $672 \times 672$ to $4032 \times 4032$. We observe that increasing the token budget $K$ at any specific scale consistently leads to an improvement in accuracy. This can be attributed to the synergistic and complementary nature of multi-scale tokens, where low-resolution tokens capture broader visual context, whereas high-resolution tokens provide essential fine-grained details. Notably, these performance gains are relatively modest and tend to plateau as $K$ increases. This suggests that our method is highly effective at identifying the most informative visual tokens even under stringent budget constraints, regardless of the input resolution.

**Resolution Robustness and Scale Trade-offs.** To evaluate the impact of input resolution, we conducted inference by downsampling 8K images to 4K. Empirical results on XLRS-Bench demonstrate that 4K inputs yield performance on par with, and in some cases superior to, the original 8K images (Table 5). Interestingly, we observe that under an identical token budget, the 4K configuration marginally outperforms the 8K baseline by approximately $0.1\%$. This phenomenon can be attributed to the increased contextual coverage per token. At lower resolutions, each token encapsulates a broader spatial area, thereby providing the model with richer global semantic cues. This suggests that for resource-constrained inference, the benefit of enhanced contextual density in 4K can outweigh the loss of extreme fine-

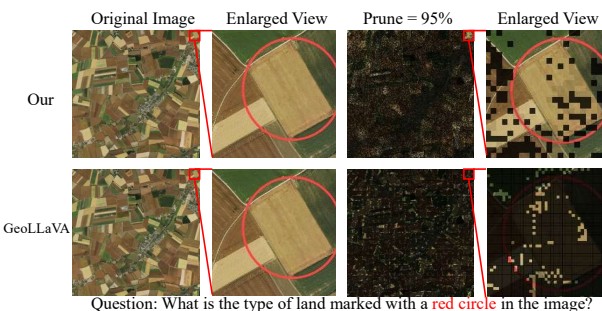

Question: What is the type of land marked with a red circle in the image?

*Figure 5.* Qualitative comparisons of our method and GeoLLaVA-8K under 95% pruning ratio.

*Table 5.* Performance analysis on XLRS-Bench under varying resolutions and token budgets.

| Kept Tokens | Image Resolution | Perception Mean | Reasoning Mean | w.Avg. |
|---|---|---|---|---|
| 3k | 8k | 40.38 | 57.83 | 43.00 |
| 4k | 8k | 40.61 | 57.52 | 43.12 |
| 3k | 4k | 40.50 | 58.04 | 43.12 |
| 4k | 4k | 40.73 | 57.83 | 43.28 |

grained pixel details in 8K. These findings underscore the robustness of our framework and its ability to maintain high accuracy without the prohibitive computational overhead associated with ultra-high-resolution inputs.

**Effect of Cluster Count ($k$).** Table 7 shows that the model is highly robust to the number of clusters, with performance fluctuations remaining within 1%. We observe that at $k = 600$, which matches the average granularity of our segmentation-based partition ($\approx 600$), the model achieves an average accuracy of 41.4%. This result is nearly identical to the peak performance, demonstrating that our clustering mechanism effectively merges redundant visual tokens and captures structural information comparable to explicit image segmentation.

**Qualitative Visualization.** Figure 5 compares our method with GeoLLAVA on remote-sensing imagery at a 95% pruning ratio. While GeoLLAVA indiscriminately discards essential information and causes severe degradation in the region of interest (red circle), our region-aware strategy consistently preserves critical semantic features. This demonstrates the robustness of our approach even under extreme pruning conditions. Such high-fidelity preservation ensures that the model retains sufficient visual cues for accurate scene interpretation and spatial reasoning.

**Visualization of Query-Guided Attention.** We visualize the text-to-image attention maps in Figure 6 to validate the reliability of our importance assessment. These visualizations demonstrate a precise spatial alignment between the textual queries and the visual activations. As evidenced in the enlarged views, the model accurately localizes the specific objects described in the questions. For instance,

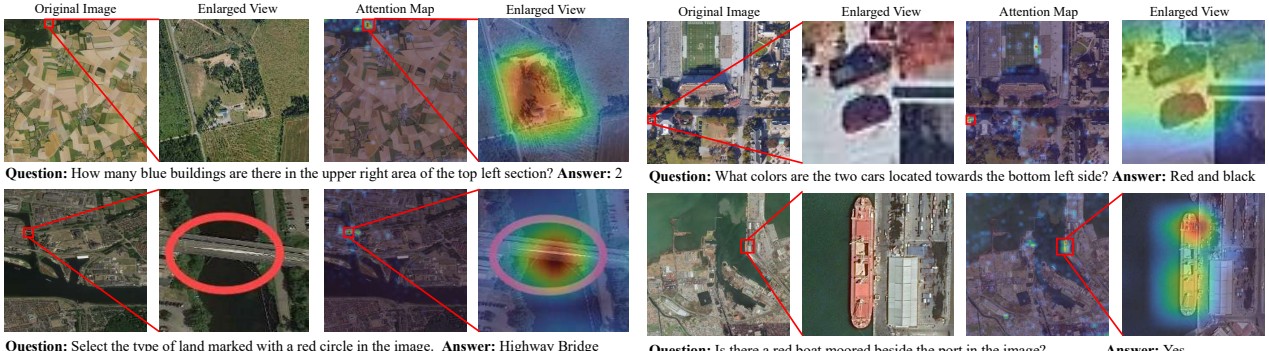

*Figure 6.* Qualitative visualization of region-aware attention on remote-sensing images. We display original images and enlarged views alongside their corresponding attention maps. The heatmaps demonstrate that the model accurately focuses on semantically relevant regions—such as specific vehicles or infrastructure—aligned with the textual query, while assigning lower importance to the background.

*Table 6.* Performance on XLRS-Bench across multiple resolutions ($672 \times 672$, $1344 \times 1344$, $2688 \times 2688$ and $4032 \times 4032$) under varying scale-specific token budgets $K$.

| Resolution | Other scales | $K$ | Perception | Reasoning | w.Avg. |
|---|---|---|---|---|---|
| 672×672 | 320,600,2400 | 0 | 40.76 | 57.83 | 43.31 |
| 672×672 | 320,600,2400 | 40 | 40.88 | 58.04 | 43.44 |
| 672×672 | 320,600,2400 | 80 | 40.95 | 57.83 | 43.47 |
| 672×672 | 320,600,2400 | 120 | 41.15 | 58.04 | 43.67 |
| 1344×1344 | 120,600,2400 | 80 | 40.69 | 57.61 | 43.21 |
| 1344×1344 | 120,600,2400 | 160 | 40.88 | 57.17 | 43.33 |
| 1344×1344 | 120,600,2400 | 240 | 41.07 | 57.39 | 43.51 |
| 1344×1344 | 120,600,2400 | 320 | 41.15 | 58.04 | 43.67 |
| 2688×2688 | 120,320,2400 | 0 | 40.80 | 58.04 | 43.38 |
| 2688×2688 | 120,320,2400 | 200 | 40.84 | 57.83 | 43.38 |
| 2688×2688 | 120,320,2400 | 400 | 41.07 | 57.83 | 43.57 |
| 2688×2688 | 120,320,2400 | 600 | 41.15 | 58.04 | 43.67 |
| 4032×4032 | 120,320,600 | 400 | 40.99 | 57.39 | 43.44 |
| 4032×4032 | 120,320,600 | 1200 | 40.99 | 57.61 | 43.47 |
| 4032×4032 | 120,320,600 | 2000 | 40.38 | 58.04 | 43.02 |
| 4032×4032 | 120,320,600 | 2800 | 41.07 | 57.83 | 43.57 |

*Table 7.* Ablation study: Accuracy (%) across Perception and Reasoning tasks under different cluster counts ($k$).

| Metric | 10 | 50 | 120 | 240 | 360 | 480 | 600 | 720 | 1000 |
|---|---|---|---|---|---|---|---|---|---|
| Perception | 37.2 | 37.6 | 37.7 | 37.3 | 37.4 | 37.3 | 37.5 | 37.1 | 37.2 |
| Reasoning | 62.4 | 64.1 | 63.5 | 62.4 | 63.3 | 63.7 | 63.9 | 63.3 | 63.0 |
| Average | 40.9 | **41.6** | **41.6** | 41.1 | 41.3 | 41.2 | 41.4 | 41.0 | 41.1 |

when querying regarding minute details like car colors or a moored boat, the attention mechanism sharply focuses on the target pixels. Simultaneously, it effectively suppresses the activations of irrelevant background areas or non-target objects. This selective activation is crucial for our method. It indicates that the cross-modal attention successfully filters out noise. Consequently, using these scores for token pruning ensures that the system retains only the critical semantic evidence required for accurate reasoning.

## 5. Conclusion

In this paper, we present UHR-BAT, a novel budget-aware token compression framework specifically engineered for ultra-high-resolution (UHR) remote sensing understanding. By integrating query-guided importance estimation with a region-wise preserve-and-merge strategy, our method effectively captures task-relevant details and reduces visual redundancy. Extensive experiments on the XLRS-Bench and RSHR-Bench demonstrate that UHR-BAT achieves state-of-the-art performance with substantially reduced token budgets, offering a superior trade-off between accuracy and efficiency. Our work provides a practical and scalable solution for the deployment of advanced MLLMs in resource-constrained geospatial applications.

## Impact Statement

UHR-BAT provides a resource-efficient solution for deploying multimodal large language models (MLLMs) in ultra-high-resolution remote sensing scenarios, even under stringent computational and context budgets. By decoupling input resolution from the quadratic explosion of visual tokens, our framework makes it feasible to process kilometer-scale imagery on commodity hardware or resource-constrained edge platforms. This democratization of UHR processing capabilities has significant implications for critical real-world applications.

## Acknowledgement

This work is supported in part by the National Natural Science Foundation of China (62576160 (W.L.), 62304104 (Y.Y.)), Leading-edge Technology Program of Jiangsu Natural Science Foundation (BK20232004 (F.M.)), the New Cornerstone Science Foundation through the XPLORER PRIZE, Fundamental Research Funds for the Central Universities (KG202514), and 111 Center (B26023).

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

# A. Background.

**Resource-Efficient Training and Inference for High-Resolution Remote Sensing.** The rapid proliferation of high-resolution remote sensing platforms, including satellites, Unmanned Aerial Vehicles (UAVs), and precision agriculture systems has significantly expanded the practical utility of aerial imagery. However, deploying state-of-the-art models in these domains is often hindered by stringent hardware constraints. Motivated by these real-world challenges, we investigate model training and inference under restricted computational budgets. Our framework enables the training of **8K ultra-high-resolution** images on a single **NVIDIA RTX A6000 (48GB)**, while maintaining a peak inference memory footprint of approximately **30GB**. To achieve this, we implement a multi-scale fixed-budget strategy, allocating $80, 320, 600$, and $2000$ tokens for input resolutions of $672 \times 672$, $1344 \times 1344$, $2688 \times 2688$, and $4032 \times 4032$, respectively. Our approach yields a practical inference latency of roughly **4 seconds** per image, striking a favorable balance between high-fidelity processing and computational feasibility. This performance underscores the potential of our model for broader integration into resource-constrained edge platforms, such as onboard UAV processing systems.

# B. Detailed Experiment Settings.

**Benchmarks and Metrics.** We conduct experiments on three widely-used benchmarks to comprehensively evaluate our model's multimodal capabilities, including general visual understanding, compositional reasoning, OCR-centric reasoning, robustness in real-world scenarios, and object hallucination behavior: XLRS-Bench (Wang et al., 2025b), MME-Realworld-RS (Fu et al., 2025), and RSHR-Bench (Dang et al., 2025c).

**XLRS-Bench.** XLRS-Bench is a comprehensive benchmark designed to evaluate multimodal perception and reasoning on *extremely large* ultra-high-resolution remote-sensing imagery. It features the largest average image size reported to date (approximately $8500 \times 8500$ pixels) with human-verified annotations, and defines 16 sub-tasks spanning diverse perceptual skills and higher-level reasoning abilities. By emphasizing complex semantic relations and spatiotemporal understanding in real-world RS scenes, XLRS-Bench provides a diagnostic testbed for a model's fine-grained recognition and multi-step decision-making on large-scale aerial imagery.

**XLRS-Bench Task Abbreviations.** We report results on the perception and reasoning dimensions using Level-3 VQA capabilities. For Perception, OC denotes Overall Counting, RC denotes Regional Counting, OLUC denotes Overall Land Use Classification, RLUC denotes Regional Land Use Classification, OCC denotes Object Classification, OCL denotes Object Color, OMS denotes Object Motion State, and OSR denotes Object Spatial Relationship. For Reasoning, RP denotes Route Planning, AD denotes Anomaly Detection and Interpretation, ECR denotes Environmental Condition Reasoning, CCR denotes Counting with Complex Reasoning, and RCCD denotes Regional Counting with Change Detection. All tasks use multiple-choice answering, where most sub-tasks adopt A/B/C/D options and OMS adopts a binary A/B format.

**MME-RealWorld.** MME-RealWorld is a large-scale, fully human-annotated benchmark targeting high-resolution, real-world visual scenarios via multiple-choice question answering across multiple domains. In this work, we adopt its Remote Sensing split (MMERealWorld-RS) to assess robustness under high-resolution overhead imagery, where models must accurately perceive small, dense objects and perform reasoning such as identification and counting in large maps. The RS split thus complements conventional VQA-style evaluations by stressing real-world difficulty induced by resolution, clutter, and long-range spatial context.

**RSHR-Bench.** RSHR-Bench is a super-high-resolution remote-sensing benchmark designed for faithful assessment of multimodal visual understanding and complex reasoning at *native* spatial resolutions. It comprises 5,329 full-scene satellite/aerial/UAV images with long side $\geq 4,000$ pixels (up to $\sim 300$ megapixels), sourced from widely used RS corpora and UAV collections. In this work, we focus on its multiple-choice VQA (MCQ) setting, which evaluates decision-making within a fixed answer space (e.g., A/B/C/D), and spans Perception, Reasoning, and Multi-turn tasks. To explicitly reduce reliance on language priors and mitigate guessing effects in the multiple-choice format, RSHR-Bench applies strong LLM adversarial filtering followed by rigorous human verification.

**RSHR-Bench Task Abbreviations.** For Perception, COL denotes Color Detection, SHP denotes Shape Recognition, DET denotes Detection, OC denotes Object Classification, REL denotes Object Spatial Relationship, OGD denotes Object Grounding, RG denotes Regional Grounding, OCN denotes Object Counting, and RCN denotes Regional Counting. For Reasoning, AD denotes Anomaly (single-turn), FP denotes Future Prediction (multi-image), MRJC denotes Multi-region

Joint Contrast (multi-image), MRJCS denotes Multi-region Joint Contrast (single-image, multi-box), and OSJ denotes Object State Judgment (single-turn).

**Baselines.** As summarized in Table 2, we compare against three baseline groups. Remote-sensing VLMs include EarthDial (Soni et al., 2025), GeoChat (Kuckreja et al., 2024), GeoLLaVA-8K (Wang et al., 2025a), and VHM (Pang et al., 2025), which are tailored to overhead imagery and geospatially grounded vision–language understanding. Open-source general-purpose VLMs cover InternVL2.5-8B (Chen et al., 2024b), InternVL3.5-8B (Wang et al., 2025c), MiniCPM2_6 (Yao et al., 2024), Phi-3.5-Vision (Abdin et al., 2024), Qwen2.5-VL-7B (Bai et al., 2025), DeepSeek-VL (Lu et al., 2024), and VILA-HD (Shi et al., 2025), representing strong off-the-shelf multimodal perception and reasoning. We also report closed-source VLMs (GPT5 (OpenAI, 2025), GPT-4o / GPT-4o-mini (Hurst et al., 2024), and Gemini-2.5-pro (Comanici et al., 2025)) as reference baselines, and for models that support high-resolution inference (e.g., GeoLLaVA-8K and VILA-HD), we follow their recommended settings to handle 4K+ remote-sensing inputs.

**More Details of Our Method.** We initialize our model based on the LLaVA-Next architecture, incorporating CLIP-ViT-L/14-336 as the vision encoder and LongVA-7B as the language backbone. The model is subsequently fine-tuned using only 10K image-text pairs sampled from SuperRS-VQA and HighRs-VQA. During fine-tuning, we utilize four different resolutions—672×672, 1344×1344, 2688×2688, and 4032×4032. Images are resized and padded to reach these target scales. We use a batch size of 16, learning rates of $1 \times 10^{-6}$ for the visual components and $5 \times 10^{-6}$ for the projection layers interacting with the LLM, and the use of ZeRO-2 parallelism with a training epoch of 2. For evaluation, baseline results are sourced from the original dataset papers. Regarding our model, in the case of XLRS-Bench, the budget $B_s$ is assigned as 180, 1320, 1600, and 8000 corresponding to the four resolutions. Similarly, in the case of MMERealWorld-RS and RSHR-Bench, we adopt budgets of 80, 320, 1600, and 4000, respectively.

# C. Implementation Details and Derivations

## C.1. Detailed Pseudocode

To efficiently reduce the sequence length while preserving local semantic structures, we introduce the Region-aware Token Pruning mechanism. As detailed in Algorithm 1, our method first partitions the feature map into disjoint semantic regions to capture spatial priors. We then assess token importance within each region and employ a preserve-and-merge strategy to retain salient information. Finally, we prioritize regions based on their average importance and truncate the sequence to satisfy a predefined computational budget $B_s$.

## C.2. Region Partition Instantiations

**Feature+coordinate K-means Partition.** Let $\mathbf{f}_i \in \mathbb{R}^D$ be the visual feature associated with token $i$ and $(x_i, y_i) \in [0, 1]^2$ be its normalized center coordinate. Distinct from natural images, remote sensing scenes often exhibit strong spatial correlation, where proximal tokens are highly likely to represent the same physical object or homogeneous region. To exploit this characteristic, we incorporate the spatial coordinates of tokens into the clustering process to identify semantic redundancy by imposing explicit spatial constraints on clustering. Specifically, we $\ell_2$-normalize the features and concatenate these scaled coordinates to form the clustering embedding $\mathbf{u}_i \in \mathbb{R}^{D+2}$:

$$\hat{\mathbf{f}}_i = \frac{\mathbf{f}_i}{\|\mathbf{f}_i\|_2}, \qquad \mathbf{u}_i = \left[\lambda_f \hat{\mathbf{f}}_i^\top, \ \lambda_{xy} x_i, \ \lambda_{xy} y_i\right]^\top. \tag{14}$$

K-means with $k$ centers iteratively partitions the input space and minimizes the objective $J$ to find centroids $\{\boldsymbol{\mu}_r\}_{r=1}^k$:

$$J = \sum_{i=1}^N \|\mathbf{u}_i - \boldsymbol{\mu}_{c_i}\|_2^2, \qquad c_i = \arg \min_{1 \le r \le k} \|\mathbf{u}_i - \boldsymbol{\mu}_r\|_2^2, \tag{15}$$

yielding clusters $\{\mathcal{S}_r\}_{r=1}^k$ as the partition $\mathcal{P}^{(s)}$. By integrating spatial distance, the clustering process effectively groups tokens that are both semantically similar and geographically close. During token selection, we merge these redundant tokens within each cluster, thereby condensing the representation while preserving the essential land-cover structures.

**SAM-induced Partition and Token Mapping.** To identify tokens that represent the same semantic information, we employ a frozen SAM model to partition the image into distinct regions. To this end, we perform a token-level mapping of the

---

**Algorithm 1** Region-aware Token Pruning

---

**Require:** Input feature map $\mathbf{X}$, Number of regions $k$, Pruning ratio $\rho$.
**Ensure:** Pruned tokens $\mathbf{X}_{out}$.
 1: **// Step 1: Region Partition**
 2: Construct clustering embeddings $\mathbf{u}_i$ for all tokens via Eq. (14).
 3: Obtain regions $\{\mathcal{S}_1, \ldots, \mathcal{S}_k\}$ by minimizing Eq. (15) (K-means) or via Eq. (16) (SAM).
 4: **// Step 2: Importance Assessment**
 5: Compute the attention score map $A_{u,v}^{(s)}$ for each scale according to Eq. (7).
 6: Derive the individual token importance scores $\{a_i^{(s)}\}_{i=1}^{N_s}$ following Eq. (8).
 7: Calculate the average importance score $\{\mu_m^{(s)}\}_{m=1}^{R_s}$ for each region as defined in Eq. (9).
 8: **// Step 3: Region-wise Preserve, Merge, and Priority Budgeting**
 9: Initialize candidate sequence $E_{\text{cand}}^{(s)} \leftarrow [\,]$
10: **for** each region $\mathcal{S}_m^{(s)} \in \mathcal{P}^{(s)}$ **do**
11: $\quad \mu_m^{(s)} \leftarrow \frac{1}{|\mathcal{S}_m^{(s)}|} \sum_{i \in \mathcal{S}_m^{(s)}} a_i^{(s)}$
12: $\quad \mathcal{K}_m^{(s)} \leftarrow \{i \in \mathcal{S}_m^{(s)} \mid a_i^{(s)} \geq \mu_m^{(s)}\}$
13: $\quad \mathcal{R}_m^{(s)} \leftarrow \mathcal{S}_m^{(s)} \setminus \mathcal{K}_m^{(s)}$
14: $\quad$ **if** $|\mathcal{R}_m^{(s)}| > 0$ **then**
15: $\quad\quad \tilde{\mathbf{e}}_m^{(s)} \leftarrow \text{MeanPool}(\{\mathbf{h}_i^{(s)} \mid i \in \mathcal{R}_m^{(s)}\})$
16: $\quad$ **end if**
17: **end for**
18: Sort region indices $\pi$ by descending average importance $\mu_m^{(s)}$
19: **for** $r = 1$ **to** $R_s$ **do**
20: $\quad m \leftarrow \pi(r)$
21: $\quad$ Sort tokens in $\mathcal{K}_m^{(s)}$ by $a_i^{(s)}$ in descending order
22: $\quad$ Append sorted tokens in $\mathcal{K}_m^{(s)}$ to $E_{\text{cand}}^{(s)}$
23: $\quad$ **if** $\tilde{\mathbf{e}}_m^{(s)}$ exists **then**
24: $\quad\quad$ Append $\tilde{\mathbf{e}}_m^{(s)}$ to $E_{\text{cand}}^{(s)}$
25: $\quad$ **end if**
26: **end for**
27: $\bar{E}^{(s)} \leftarrow E_{\text{cand}}^{(s)}[1 : B_s]$ {Truncate to budget $B_s$}
28: **return** $\bar{E}^{(s)}$

---

pixel-level segmentation results. Specifically, given pixel-level labels from a segmentation model (e.g., SAM (Kirillov et al., 2023)), we map them to token regions by majority vote over the pixel set $\Omega_i$ covered by token $i$:

$$\text{label}_{\text{token}}(i) = \arg\max_c \sum_{(x,y) \in \Omega_i} \mathbb{I}\big[\text{label}(x, y) = c\big]. \tag{16}$$

Tokens with the same token-level label form a region set $\mathcal{S}_m$, thereby effectively producing the final spatial partition $\mathcal{P}^{(s)}$. Ties encountered in this process are broken deterministically (e.g., by choosing the smallest label id).

### C.3. Spatial Continuity and Bilinear Interpolation

To bridge the resolution discrepancy between the coarse importance scores and the target high-resolution feature maps, we employ bilinear interpolation to derive spatially aligned importance values.

Formally, let $G$ be a discrete grid-valued map representing the low-resolution importance scores. For any continuous coordinate $(x, y) \in \mathbb{R}^2$ on the target high-resolution grid, we define its four surrounding integer neighbors on the source grid as $x_0 = \lfloor x \rfloor, x_1 = x_0 + 1, y_0 = \lfloor y \rfloor$, and $y_1 = y_0 + 1$.

Let $Q_{mn} = G(x_m, y_n)$ denote the sampled importance values at these grid corners for $m, n \in \{0, 1\}$. The interpolation

operator $\Psi$ computes the refined importance value at the high-resolution coordinate as:

$$\Psi(G, (x, y)) = Q_{00}(x_1 - x)(y_1 - y) + Q_{10}(x - x_0)(y_1 - y) \\ + Q_{01}(x_1 - x)(y - y_0) + Q_{11}(x - x_0)(y - y_0). \tag{17}$$

### C.4. Basic Properties of the Preserve-and-Merge Rule

**Lemma 1 (Structural Coverage and Representation Bound).** For any given semantically coherent region $\mathcal{S}_m^{(s)}$ with intrinsic feature variance $\sigma_m^2 = \frac{1}{|\mathcal{S}_m^{(s)}|} \sum_{i \in \mathcal{S}_m^{(s)}} \|h_i^{(s)} - \bar{h}_m^{(s)}\|^2$, the proposed preserve and merge rule theoretically ensures that each selected region contributes at least one high salience token such that the retained set defined in Eq. (10) is guaranteed to be non-empty, i.e., $\mathcal{K}_m^{(s)} \neq \emptyset$, while simultaneously, the local reconstruction error $\epsilon_m$ incurred by the merged tokens is strictly bounded by the intra region feature dispersion.

**Proof.** The non empty property follows from the definition of the regional mean $\mu_m^{(s)}$: if $\forall i \in \mathcal{S}_m^{(s)}, a_i^{(s)} < \mu_m^{(s)}$, then $\sum a_i^{(s)} < |\mathcal{S}_m^{(s)}| \mu_m^{(s)}$, which contradicts the formulation of average importance. This ensures that critical evidence in every partitioned sector is prioritized at the algorithmic level rather than being marginalized by global noise.

To prove the error bound, let the compressed representation of $\mathcal{S}_m^{(s)}$ be $\hat{E}_m = \{h_i^{(s)} : i \in \mathcal{K}_m^{(s)}\} \cup \{\tilde{e}_m^{(s)}\}$. Since $\tilde{e}_m^{(s)}$ is the centroid of the merged subset $\mathcal{R}_m^{(s)}$, the local reconstruction error $\epsilon_m$ (measured by the sum of squared distances) is:

$$\epsilon_m = \sum_{i \in \mathcal{R}_m^{(s)}} \|h_i^{(s)} - \tilde{e}_m^{(s)}\|^2 = |\mathcal{R}_m^{(s)}| \cdot \mathrm{Var}(\mathcal{R}_m^{(s)}) \tag{18}$$

Because the region $\mathcal{S}_m^{(s)}$ is constructed via semantic clustering, tokens within $\mathcal{S}_m^{(s)}$ exhibit high homogeneity. By selectively preserving high salience tokens $\mathcal{K}_m^{(s)}$ and merging the remaining redundant background tokens into their local centroid, we minimize the semantic shift. The total error is thus strictly constrained by the compactness of the semantic partition, ensuring that the compressed sequence remains region faithful even under extreme budget constraints.

**Lemma 2 (First-moment preservation on the merged subset).** For any region with a non-empty merging set $|\mathcal{R}_m^{(s)}| > 0$, the merged token $\tilde{\mathbf{e}}_m^{(s)}$ defined in Eq. (11) preserves the first moment (the centroid) of the features within $\mathcal{R}_m^{(s)}$:

$$\sum_{i \in \mathcal{R}_m^{(s)}} \mathbf{h}_i^{(s)} = |\mathcal{R}_m^{(s)}| \tilde{\mathbf{e}}_m^{(s)}. \tag{19}$$

**Proof.** By definition, the merged representation is the arithmetic mean of the features in the merging set: $\tilde{\mathbf{e}}_m^{(s)} = \frac{1}{|\mathcal{R}_m^{(s)}|} \sum_{i \in \mathcal{R}_m^{(s)}} \mathbf{h}_i^{(s)}$. Multiplying both sides by the cardinality $|\mathcal{R}_m^{(s)}|$ directly yields Eq. (19), confirming that the global feature sum is invariant under the merging operation.

**Robustness to Region Partitioning Methods.** To demonstrate that our framework is agnostic to the specific choice of region partitioning strategy, we evaluate its performance across diverse algorithms, including both clustering-based methods and the Segment Anything Model (SAM). Table 8 compares the results of k-means, BIRCH, and SAM on XLRS-Bench across various computational budgets. The empirical results indicate that the choice of partitioning technique has no significant impact on the final performance, with the highly efficient clustering methods achieving results comparable to the much more computationally intensive SAM. This consistency highlights the robustness of our approach and underscores its flexibility in utilizing different spatial priors without compromising accuracy.

While our framework can utilize either the Segment Anything Model (SAM) or clustering-based methods for region partitioning, we prioritize the latter in practical applications due to the prohibitive preprocessing latency associated with SAM. This efficiency gain enables our approach to be seamlessly extended to a broader range of latency-critical scenarios, particularly those requiring real-time inference and high-throughput processing.

**Impact of Context Length and Resolution Efficiency.** Table 9 ablates the performance of various MLLMs across different context lengths on XLRS-Bench. While scaling from 2K to 4K tokens yields substantial gains (e.g., Qwen2.5VL-7B improves from 38.08% to 40.58%), further extending the context to 8K offers negligible improvement or even slight

*Table 8.* Evaluation of various region partitioning methods.

| Method | 4k | 5k | 6k | 8k | 1w |
|--------|------|------|------|------|------|
| BIRCH | 43.05 | 43.28 | 43.28 | 43.31 | 43.51 |
| K-means | 43.34 | 43.57 | 41.43 | 43.73 | 43.31 |
| SAM | 43.28 | 43.54 | 43.54 | 43.57 | 43.67 |

*Table 9.* Performance on XLRS-Bench across different resolutions.

| Context | Qwen2.5VL-7B | InternVL-3.5 | Phi-3.5 |
|---------|--------------|--------------|---------|
| 2K | **38.08** | 38.41 | 37.08 |
| 4K | **40.58** | 38.60 | 37.11 |
| 8K | **40.36** | 39.80 | 37.90 |

regression (40.36%). This plateau indicates that the semantic information density required for XLRS-Bench is effectively saturated at the 4K resolution level. Consequently, processing 8K native inputs within a compressed 4K context window preserves critical visual details while significantly reducing computational overhead. Based on this observation, we adopt the 4K setting as the optimal operating point, prioritizing token efficiency without compromising downstream accuracy.

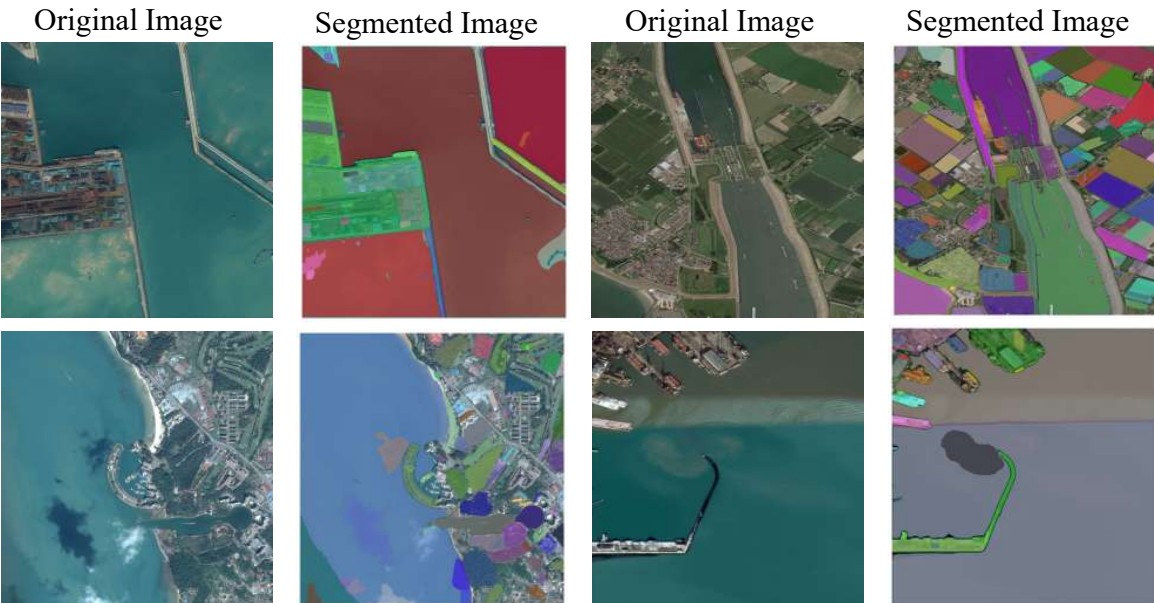

| Original Image | Segmented Image | Original Image | Segmented Image |

*Figure 7.* Qualitative examples of coarse segmentation on remote-sensing images produced by our segmentation module, used to form semantically coherent regions for region-aware processing.

**Visualization of Region Partition.** We provide qualitative examples of the region partition in Figure 7 to illustrate the intermediate results of our framework. The primary goal of this module is not to generate pixel-level accurate semantic masks, but rather to decompose the high-resolution image into distinct spatial groups. As shown in Figure 7, the SAM-based method divides the image into multiple local regions based on feature similarity. Qualitative inspection reveals that the resulting partitions generally exhibit high visual homogeneity, effectively encapsulating consistent semantic information within each region. Conversely, to account for the background which often encompasses diverse and heterogeneous visual constituents, we treat each token within the background as an independent mask category. This partition provides a structural basis for the subsequent token compression step, allowing us to process the image in a region-wise manner instead of a global manner. Even if the segmentation boundaries are coarse or contain noise, the partition successfully separates the large background areas from the complex foreground details. This spatial grouping ensures that our method can evaluate token importance within each local scope; consequently, we can preserve the diversity of the visual information across the image and avoid the domination of a single large object, regardless of the precise quality of the segmentation.

**t-SNE Visualization of K-means Clustering Results.** To further investigate the rationale behind our fixed-budget compression, we employ t-SNE to project the high-dimensional visual tokens into a 2D plane (see Figure 8). While the projection exhibits a degree of proximity between clusters—inherent to the complexity of high-resolution remote sensing manifolds—it is evident that tokens belonging to the same cluster (denoted by color) are significantly aggregated in the latent space. This *semantic proximity* confirms that a substantial number of tokens convey nearly identical visual information. By

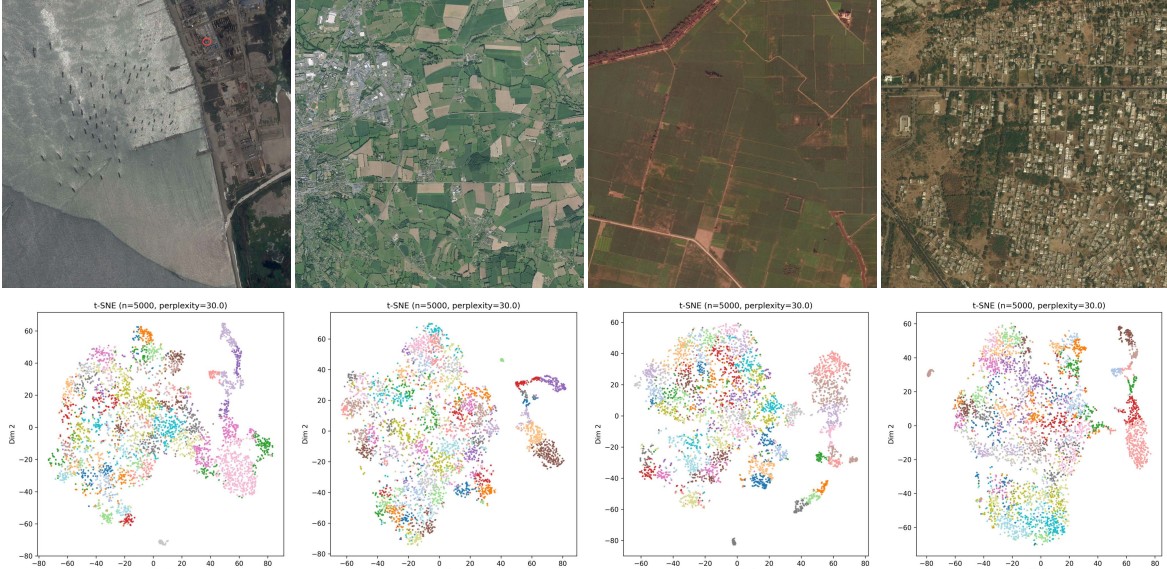

*Figure 8.* t-SNE visualization of visual tokens. Each color represents a cluster assigned by our region-based grouping module. Although the high-dimensional nature of tokens leads to some spatial overlap in the 2D projection, tokens with the same semantic labels exhibit clear clustering behavior, validating the redundancy of visual information and the feasibility of our token merging strategy.

identifying and merging these redundant tokens within each semantic cluster, we can drastically reduce the computational overhead without sacrificing critical structural details, thereby enabling efficient inference under a fixed-budget constraint.

**Visualization of Token Pruning.** We visualize the spatial distribution of retained tokens in Figure 9 and Figure 10 to demonstrate the effectiveness of our compression strategy. The figures illustrate the surviving visual tokens at varying pruning ratios, ranging from 85% to 97.5%. Figure 9 displays the results using the attention-based selection, while Figure 10 shows the results based on the clustering partition. In both scenarios, we observe that the method consistently preserves the semantic regions relevant to the user query. As the pruning ratio increases, the irrelevant background areas are progressively removed (shown as black). However, the critical visual evidence—such as specific ships, text on the ground, or swimming pools—remains visible even under extreme compression rates. This visualization confirms that our approach successfully reduces computational redundancy while maintaining the necessary information for correct reasoning.

**Qualitative Comparison and Analysis.** We present a qualitative evaluation of our method in comparison with state-of-the-art Multimodal Large Language Models, including InternVL3.5-8B, GeoLLaVA-8K, and Qwen2.5-VL-72B. The comparison results are illustrated in Figure 11. These examples cover a range of challenging tasks such as fine-grained object counting (e.g., oil tanks, swimming pools), small object attribute recognition, and spatial reasoning. As observed in the examples, baseline models often struggle with the high-resolution nature of remote sensing imagery; they frequently suffer from hallucinations or fail to recognize small objects, leading to incorrect counts. In contrast, our method successfully identifies and reasons about these minute details. This verifies that our region-aware token pruning strategy effectively retains the informative visual cues necessary for accurate answering. Furthermore, we provide additional qualitative examples in Figure 12 to demonstrate the robustness and versatility of our model across various scenarios.

**Additional Qualitative Results.** We present additional qualitative examples in Figure 12 to further demonstrate the versatility of our method. These examples cover a wide range of tasks, including spatial reasoning, fine grained attribute recognition, and scene classification. As shown in the figures, our model effectively handles high resolution inputs and accurately answers questions regarding small targets, such as identifying the status of a bridge or the color of a chimney. This confirms that our approach maintains strong performance across diverse semantic categories and complex visual scenes.

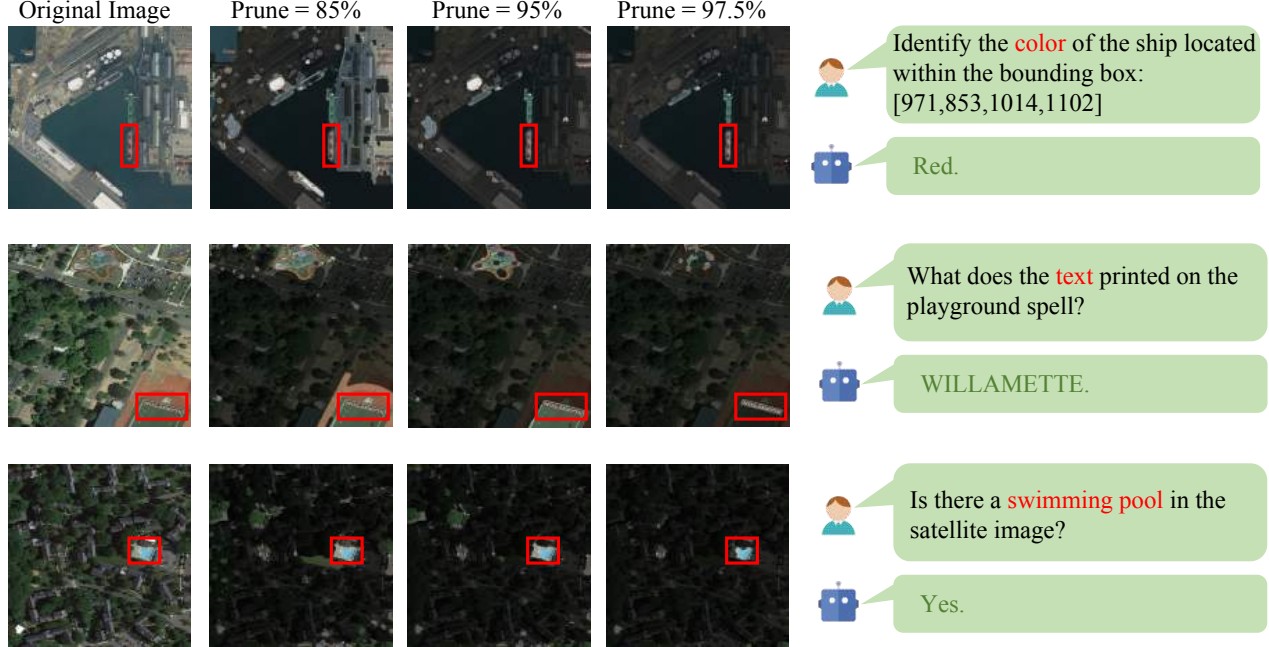

*Figure 9.* Visualization of attention-guided token pruning. We display the original image and the retained visual tokens at increasing pruning ratios. The method effectively focuses on high-attention areas, preserving query-relevant details (red boxes) like specific colors or text, while discarding the non-informative background.

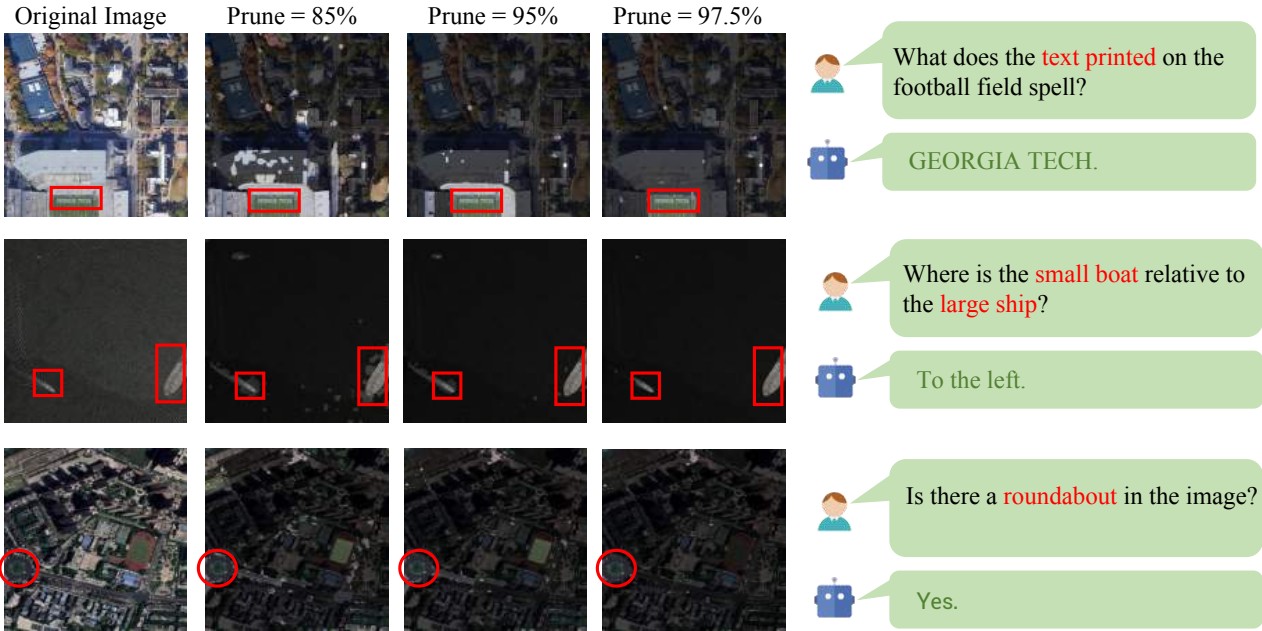

*Figure 10.* Visualization of clustering-guided token pruning. This example demonstrates how region-based partitioning helps retain structural objects. Even at a 97.5% pruning ratio, small targets like boats or roundabouts are preserved within their clusters, ensuring the model retains sufficient context for spatial reasoning.

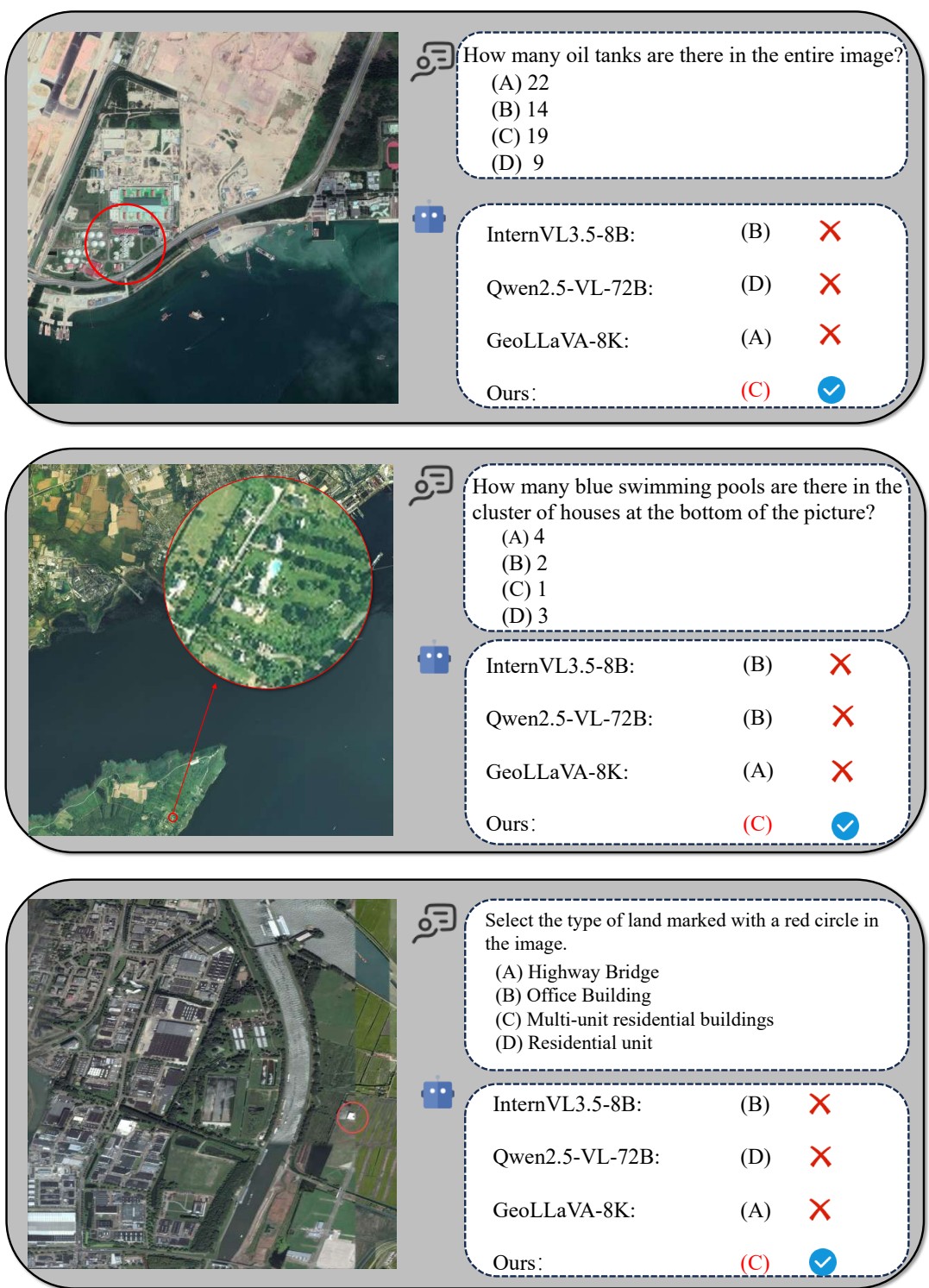

*Figure 11.* Qualitative comparison of model responses across InternVL3.5-8B, Qwen2.5-VL-72B, and our method. The examples demonstrate that our model correctly handles fine grained tasks such as counting dense oil tanks or identifying specific land types.

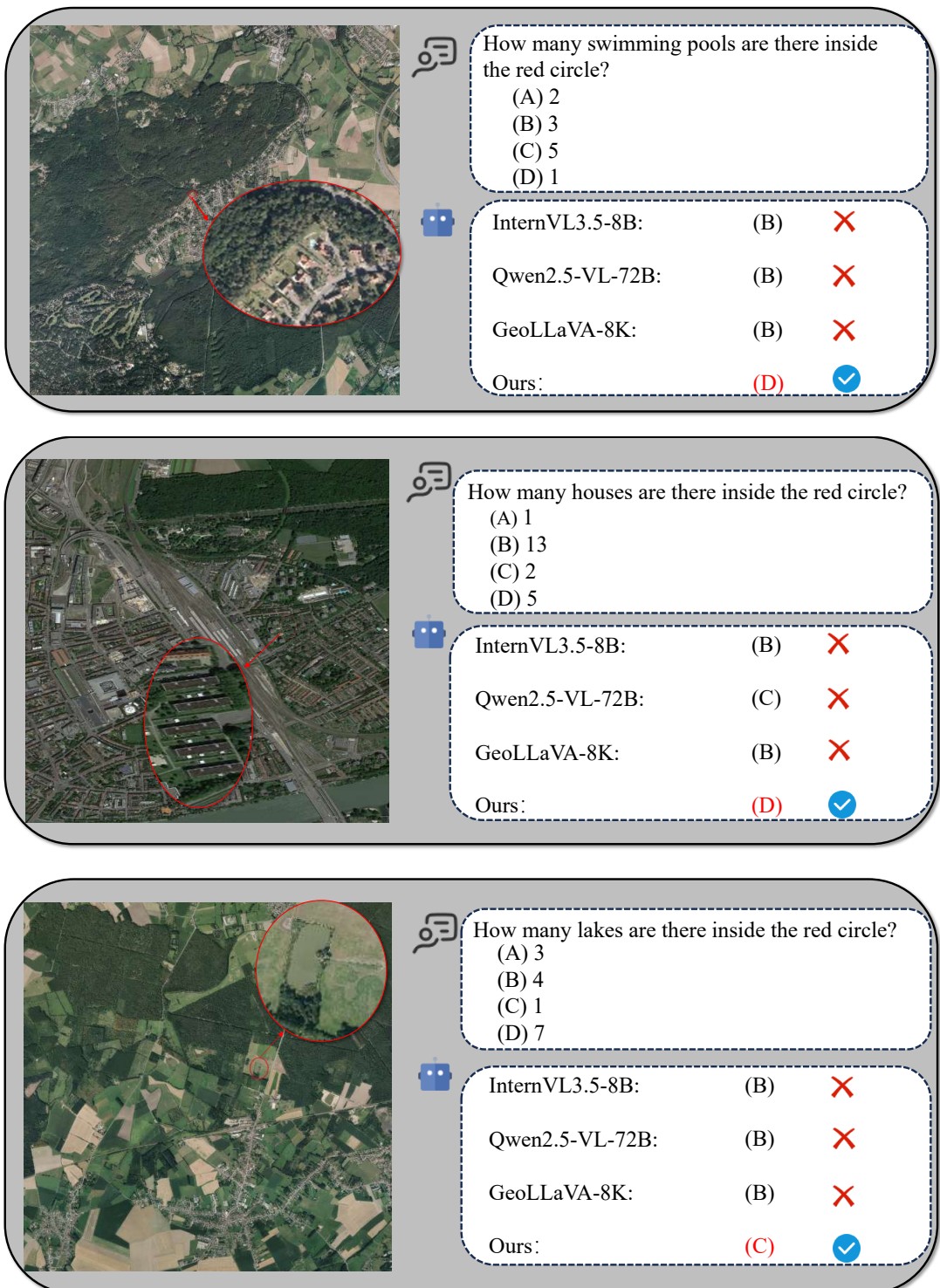

*Figure 11.* Qualitative comparison of model responses. Our method consistently provides accurate answers for challenging counting tasks within localized regions, effectively distinguishing targets like swimming pools or houses from complex backgrounds.

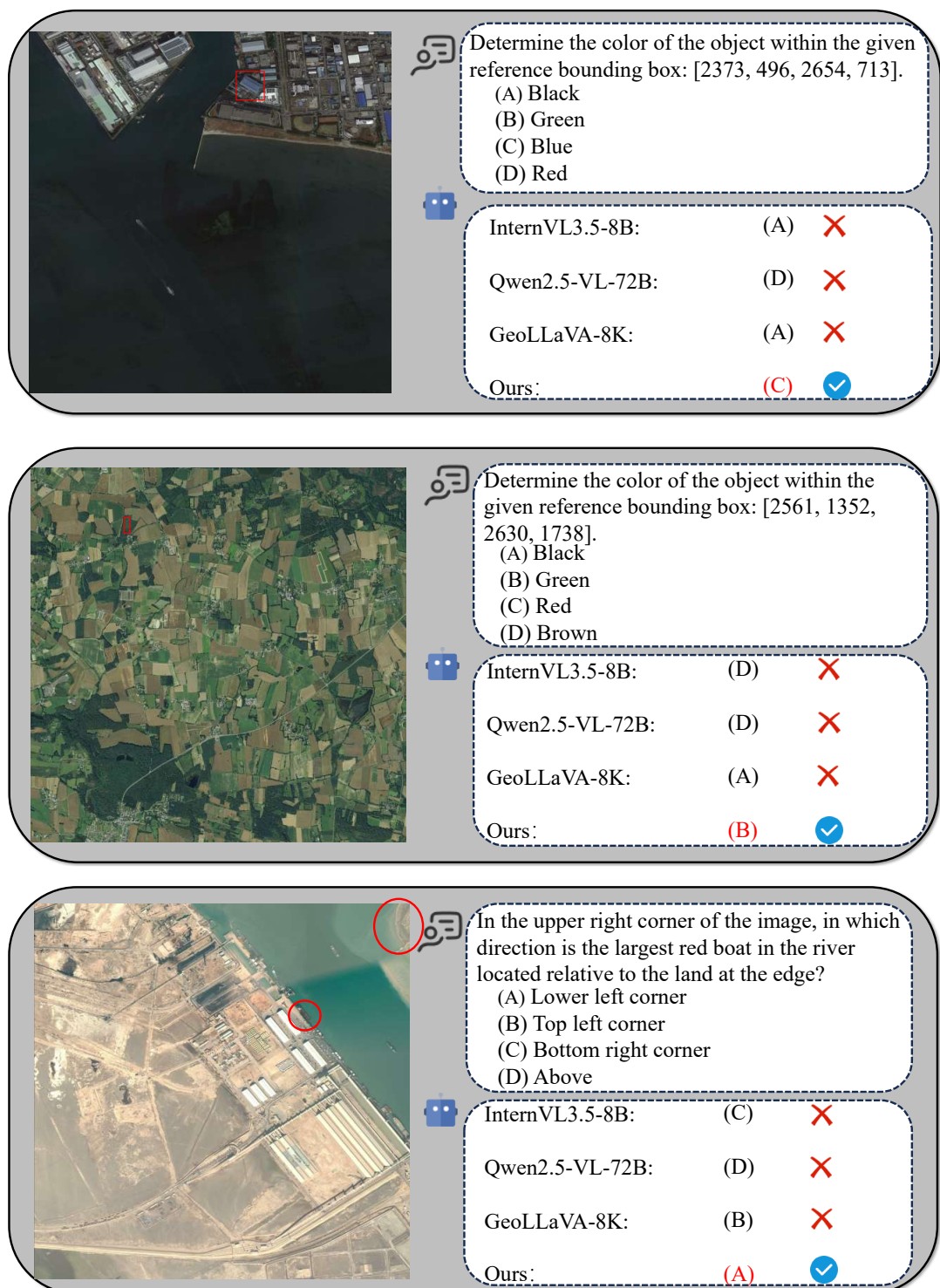

*Figure 11.* Qualitative comparison of model responses. The visualization highlights the ability of our model to ground the correct visual evidence for fine grained queries, specifically including color identification of small objects and reasoning about spatial directions.

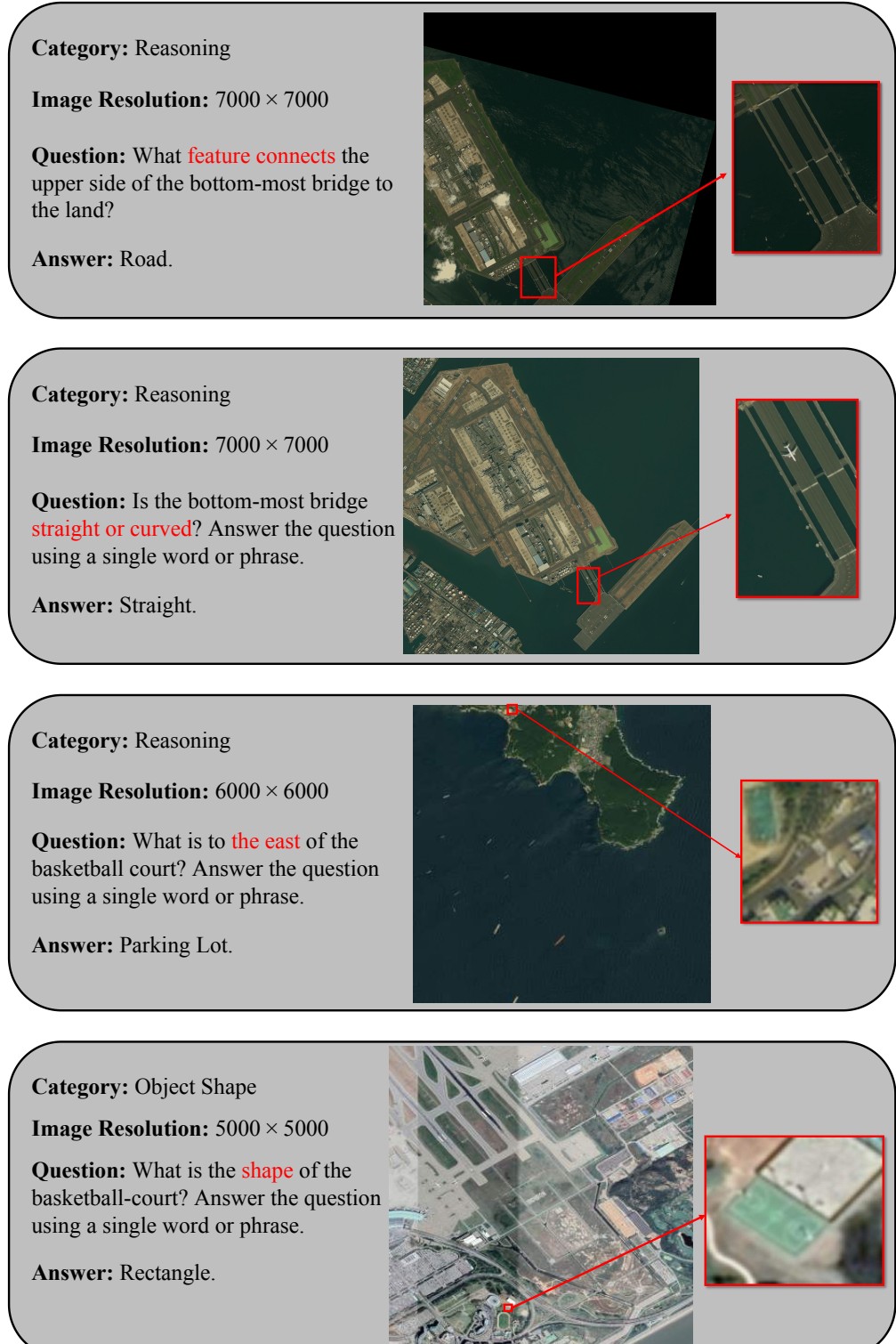

*Figure 12.* Qualitative examples of responses generated by our model. These samples illustrate the versatility of our method in handling diverse tasks, including spatial reasoning about road orientation and fine-grained attribute recognition of small objects.

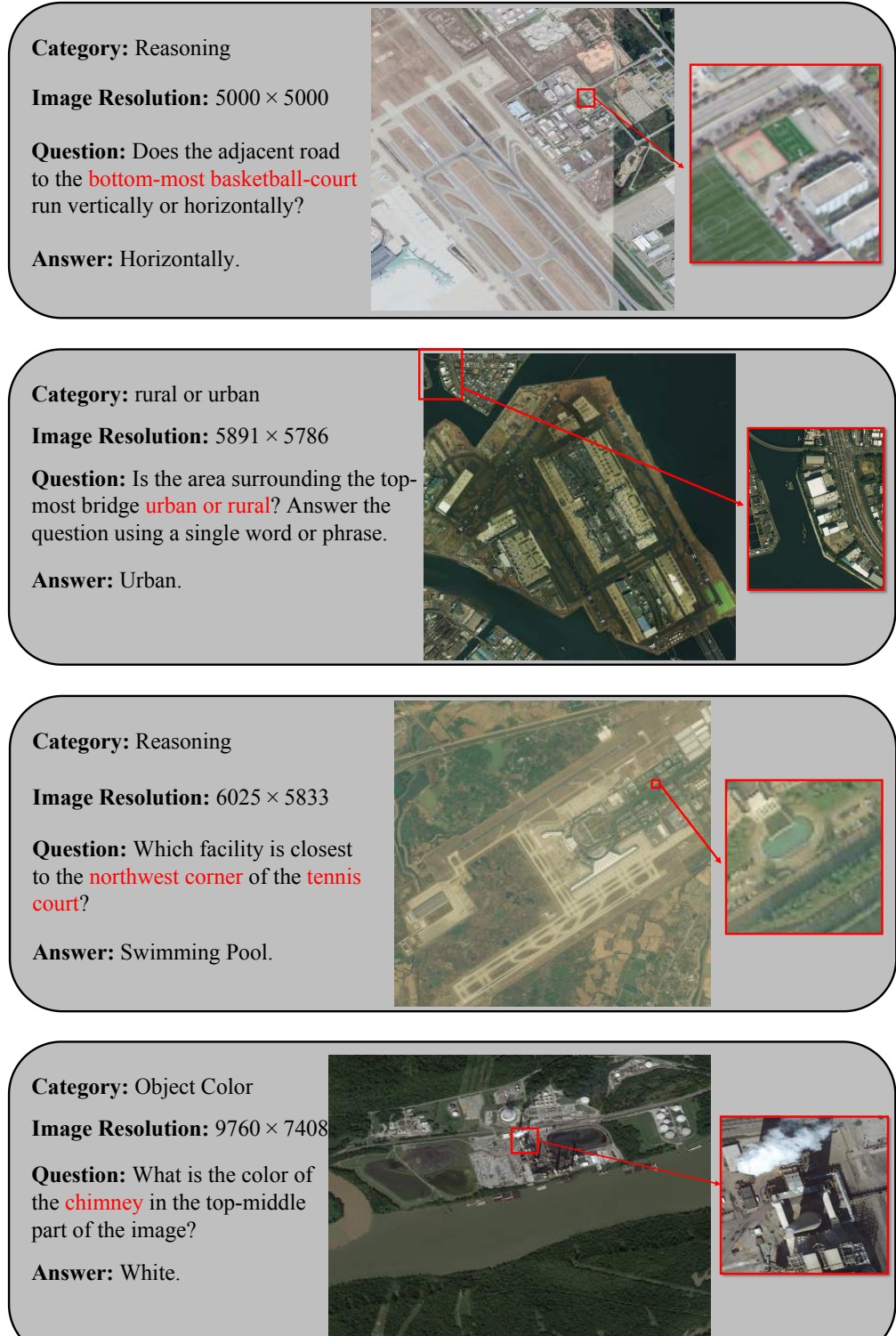

**Category:** Reasoning

**Image Resolution:** 5000 × 5000

**Question:** Does the adjacent road to the bottom-most basketball-court run vertically or horizontally?

**Answer:** Horizontally.

**Category:** rural or urban

**Image Resolution:** 5891 × 5786

**Question:** Is the area surrounding the top-most bridge urban or rural? Answer the question using a single word or phrase.

**Answer:** Urban.

**Category:** Reasoning

**Image Resolution:** 6025 × 5833

**Question:** Which facility is closest to the northwest corner of the tennis court?

**Answer:** Swimming Pool.

**Category:** Object Color

**Image Resolution:** 9760 × 7408

**Question:** What is the color of the chimney in the top-middle part of the image?

**Answer:** White.

*Figure 12.* Qualitative examples of responses generated by our model. The visualization demonstrates the ability of the model to interpret geometric shapes and complex spatial relationships, such as identifying the curvature of a bridge or determining the relative position.

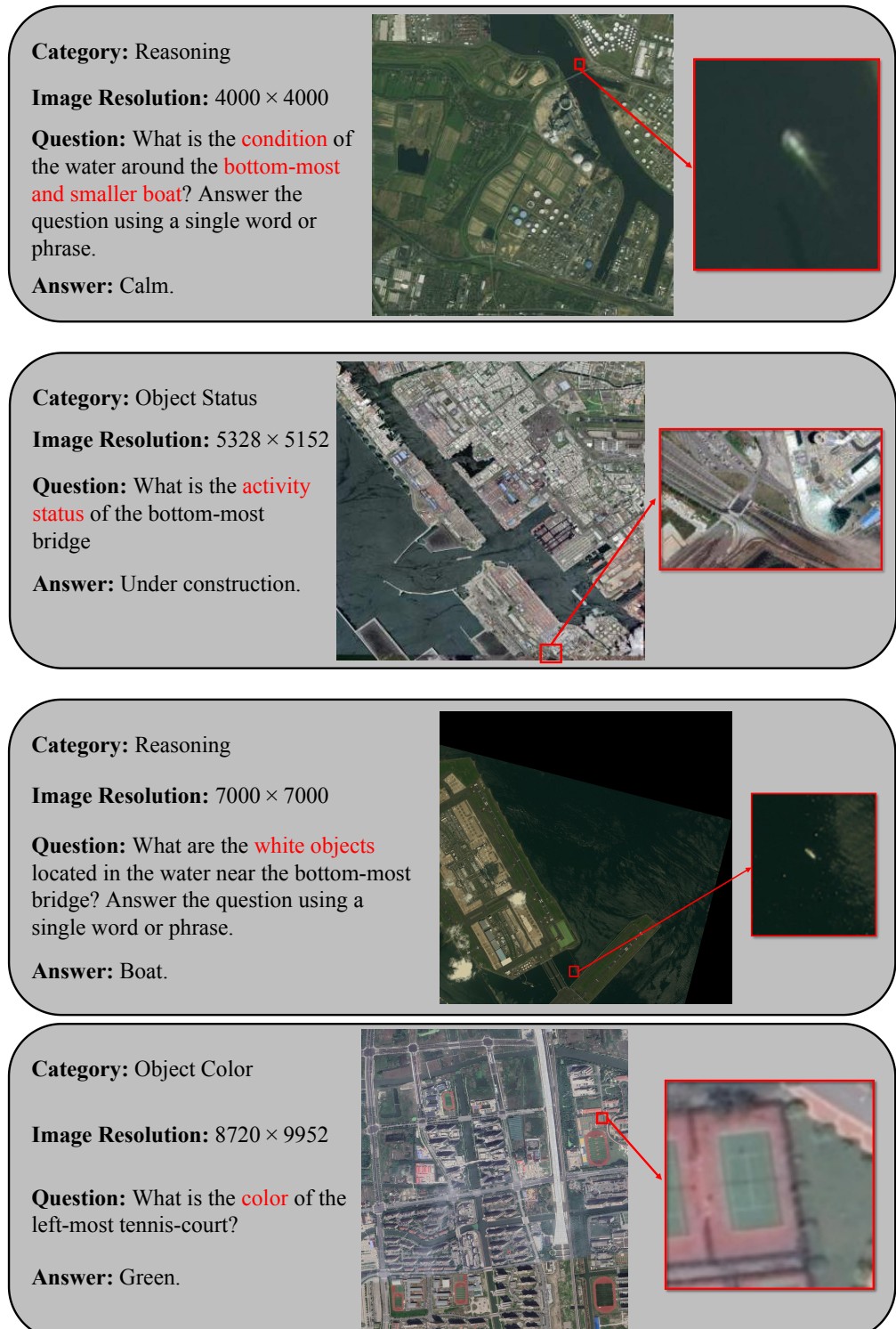

**Category:** Reasoning

**Image Resolution:** 4000 × 4000

**Question:** What is the condition of the water around the bottom-most and smaller boat? Answer the question using a single word or phrase.

**Answer:** Calm.

**Category:** Object Status

**Image Resolution:** 5328 × 5152

**Question:** What is the activity status of the bottom-most bridge

**Answer:** Under construction.

**Category:** Reasoning

**Image Resolution:** 7000 × 7000

**Question:** What are the white objects located in the water near the bottom-most bridge? Answer the question using a single word or phrase.

**Answer:** Boat.

**Category:** Object Color

**Image Resolution:** 8720 × 9952

**Question:** What is the color of the left-most tennis-court?

**Answer:** Green.

*Figure 12.* Qualitative examples of responses generated by our model. These results show that our model effectively captures dynamic scene details, such as assessing water conditions, identifying construction activities, detecting minute objects, and recognizing colors.

