# OpenReview forum: "UHR-BAT: Budget-Aware Token Compression Vision-Language model for Ultra-High-Resolution Remote Sensing"
_ICML.cc/2026/Conference — ICML 2026 regular_

### Official Review · Reviewer_WnTy · 2026-02-25

**Soundness:** 2
**Presentation:** 2
**Significance:** 2
**Originality:** 2
**Overall Recommendation:** 3
**Confidence:** 3

**Summary:**

This paper proposes a token compression method to adapt Vision-Language Model for ultra-high-resolution remote sensing imagery. The target of UHR-BAT is to allocate more resources on the regions of interest (RoI), while maintaining region faithfulness so that correlated context and minor critical targets can be preserved. To this end, UHR-BAT uses a top-down query-guided method to distinguish RoI, and then exploits Region-wise Preserve and Merge to enhance region faithfulness. Experimental results demonstrate improvements on efficiency-accuracy trade-off.

**Compliance With Llm Reviewing Policy:**

Affirmed.

**Final Justification:**

I appreciate authors' detailed responses. Most technical concerns (W1 and W2) have been addressed, but I still believe the scope of this work is somewhat narrow, which may limit its impact and contributions. Therefore, I will maintain my initial slightly negative rating, but I would not mind the paper being accepted either.

**Key Questions For Authors:**

Please refer to the weaknesses.

**Limitations:**

yes

**Strengths And Weaknesses:**

**Strengths**

1. This paper points out two important issues in token compression for ultra-high-resolution remote sensing, which are RoI detection and region faithfulness. Then, the authors propose reasonable solutions to address these issues accordingly.

2. The paper is clear, well-written, and easy to follow.

**Weaknesses**
1. Although UHR-BAT is proven to outperform various base VLMs, the experiment section lacks a comparison with other general-purpose or remote-sensing-oriented token pruning methods (e.g., [1,2]). From this perspective, it is difficult to verify the contribution of UHR-BAT.

2. Table 4 shows that UHR-BAT reduces the inference latency by around 1.6 seconds. Although it may occupy a large noticeable proportion (27.5%), the improvement itself is actually marginal from the perspective of practical use.

3. This paper concentrates on analyzing remote sensing imagery, which is probably out of the scope of ICML. Token pruning for ultra-high-resolution image is not a quite important topic in the ML community. This script is probably more suitable for a remote sensing venue.

**References**

[1] Luo, Junwei, et al. "When large vision-language model meets large remote sensing imagery: Coarse-to-fine text-guided token pruning." ICCV 2025.

[2] Zhong, Yiwu, et al. "Aim: Adaptive inference of multi-modal llms via token merging and pruning." ICCV 2025.

---

> ### Author Rebuttal · Authors · 2026-03-31
>
> We thank the **reviewer WnTy’s** positive feedback on the clarity and flow of our manuscript.
> Below, we respond point by point to the weaknesses and key questions.
> ***
> Q1: The experiments lack comparisons with general-purpose or remote-sensing-oriented token pruning methods (e.g., [1,2]).
>
> A1: We sincerely thank the reviewer for pointing out this important aspect. To address this, we additionally compare UHR-BAT with **five representative token pruning methods**: ToMe, PiToMe, ToFu, AIM, and RFM+DIP.
> As shown in Table 1, UHR-BAT consistently **outperforms all evaluated token pruning methods**.
> We will include these results and the corresponding analysis in the experiment section of the our paper.
>
> Table1. Performance comparison of UHR-BAT and other representative token pruning methods.
> |Method Category|Preserved Tokens|Method|Perception Acc.| Reasoning Acc.|Overall Acc.|
> |:-|:-|:-|:-|:-|:-|
> |General-purpose|11k|AIM([2])|39.2(-2.3)|57.4(-0.4)|42.0(-2.0)|
> |General-purpose|28k|ToMe([3])|37.7(-3.8)|53.4(-4.4)|40.0(-4.0)|
> |General-purpose|28k|PiToMe([4])|37.7(-3.8)|53.9(-3.9)|40.1(-3.9)|
> |General-purpose|28k|ToFu([5])|38.0(-3.5)|50.2(-7.6)|39.8(-4.2)|
> |Remote-sensing|Adaptive|RFM+DIP([1])|31.7(-9.8)|54.3(-3.5)|35.0(-9.0)|
> |Ours|11k|UHR-BAT|41.5|57.8|**44.0**|
> ***
> Q2: Table 4 shows only about 1.6s latency reduction, which seems marginal in practice.
>
> A2: We would like to apologize for the confusion caused by our presentation of Table 4. To clarify, Table 4 was only designed to ablate the effects of different pruning ratios within our UHR-BAT framework.
>
> To demonstrate the **true practical significance and massive absolute efficiency improvements of UHR-BAT**, we have conducted a comprehensive comparison on an NVIDIA A100 (80GB) GPU. We compared UHR-BAT against the unpruned baseline and Qwen2.5-VL-7B.
>
> The result shows that UHR-BAT demonstrates profound practical value, as we not only **overcome the Out-Of-Memory limitations of the unpruned baseline**, but also greatly cut the inference latency, FLOPs and memory usage compared to Qwen2.5-VL-7B.
>
> Table2. Comprehensive efficiency comparison of UHR-BAT against the unpruned baseline and general VLM.
> |Model|Latency(s)|TFLOPs|Memory Usage(GB)|
> |:-|:-|:-|:-|
> |Qwen2.5-VL-7B|21.75|293.52|31.96|
> |Unpruned Baseline|~180.00*|~8650.00*|~128.85* (OOM)|
> |**Ours (UHR-BAT)**|**9.04** (~19.9x faster)| **272.65** (↓ 96.8%)|**24.59** (↓ 80.9%)|
> ***
> Q3: This paper concentrates on analyzing remote sensing imagery, which is probably out of the scope of ICML. Token pruning for ultra-high-resolution image is not a quite important topic in the ML community. This script is probably more suitable for a remote sensing venue.
>
> A3: (1) We respectfully clarify that our paper is not a remote-sensing application paper, but a **general token compression method for foundation MLLMs under strict token budgets**. The core challenge we address is broadly relevant to ML: how to preserve **query-critical, multi-scale visual evidence under limited tokens**.
>
> (2) We evaluate on remote sensing because it serves as a particularly strong **stress test** for this problem. Ultra-high-resolution remote sensing images contain **sparse small objects, large backgrounds, multi-scale structure, and long-range spatial dependencies**, which expose the weaknesses of existing token pruning methods more clearly than ordinary benchmarks. At the same time, our method itself is domain-agnostic: it operates on general visual tokens and cross-modal relevance, without relying on remote-sensing-specific labels. Its key components (**query-aware scoring, cross-scale alignment, and preserve-and-merge under a budget**) are generic algorithmic choices rather than domain-specific designs.
>
> (3) To further show that the gains do not come only from remote-sensing adaptation, we report a controlled baseline trained with the same fine-tuning data but without our method (Table 3). The improvement therefore cannot be explained by domain adaptation alone. We agree that our current empirical validation is centered on remote sensing, and in the revision we will position the paper more precisely as a **general token compression framework evaluated on a challenging ultra-high-resolution benchmark**.
>
> Table3. Controlled analysis of domain adaptation vs. our method on XLRS-Bench
> |Model|Perception Acc.|Reasoning Acc.|Overall|
> |:-|:-|:-|:-|
> |LLaVA-Next (Baseline)|37.2|54.5|39.8|
> |+ RS Fine-tuning Data|38.3|57.6|41.2(+1.4)|
> |+ RS Fine-tuning Data + Our Method|41.5|57.8|**44.0(+4.2)**|
>
> Note: Baseline values are estimated because it does not fit on an NVIDIA A100 (80GB) GPU, while all other runtime, memory, and FLOPs are measured on the same GPU.
> #### References
>
> [3] Bolya, D., Fu, C.-Y., Dai, X., Zhang, P., & Feichtenhofer, C. (2023). Token Merging: Your ViT but Faster.
>
> [4] Tran, H.-C., et al. (2024). Accelerate Transformer with Spectrum Preserving Token Merging.
>
> [5] Kim, S., et al. (2023). ToFu: Token Fusion for Efficient Vision Transformers.

---

> > ### Author Rebuttal · Reviewer_WnTy · 2026-04-04
> >
> > I appreciate authors' detailed responses. Most technical concerns (W1 and W2) have been addressed, but I still believe the scope of this work is somewhat narrow, which may limit its impact and contributions. Therefore, I will maintain my initial slightly negative rating, but I would not mind the paper being accepted either.

---

> > > ### Author Response · Authors · 2026-04-07
> > >
> > > We thank the **reviewer WnTy** for this important comment. We agree that, as currently presented, the empirical study is centered on the remote-sensing setting, which may make the scope appear narrower than intended; while we chose this setting because it **makes the problem substantially more challenging**, due to sparse small objects and large background regions. In the revision, we will therefore temper the scope claim in the paper and **add additional cross-scenario evidence** to better assess whether the proposed design transfers beyond the original setting.
> > >
> > > More specifically, we conducted an additional high-resolution transfer study using **the same backbone, architectural design, token-budget setting, and evaluation protocol** as in the main paper, without introducing any domain-specific modules. In addition, following prior high-resolution visual reasoning practice[3], we further performed instruction tuning on the public **TreeVGR-SFT** dataset, while not tuning on any of the evaluation benchmarks themselves.
> > >
> > > We then evaluated the resulting model on three additional high-resolution benchmarks with broader real-world coverage: **MME-RealWorld-Lite**, **HR-Bench-8K**, and **TreeBench**. We also added a **random token selection** ablation on **MME-RealWorld-Lite** under the same token budget. Its substantially lower performance provides supporting evidence that the improvement comes from our proposed token selection strategy. Across these evaluations, the method remained consistently competitive, suggesting that the proposed design is **not tied exclusively to the original remote-sensing setting**, but can also transfer to other challenging high-resolution visual understanding settings under the same token-budget constraints.
> > >
> > > Table1. Results on MME-RealWorld-Lite[1].
> > >
> > > |Method|LLM|Perc. OCR|Perc. RS|Perc. DT|Perc. MO|Perc. AD|Reas. OCR|Reas. DT|Reas. MO|Reas. AD|Overall Avg.|
> > > |:--|:--:|--:|--:|--:|--:|--:|--:|--:|--:|--:|--:|
> > > |GPT-4o|-|81|45|65|34|37|72|50|42|33|46.4|
> > > |GPT-4o-mini|-|70|23|62|19|34|57|39|19|35|37.4|
> > > |Qwen2-VL|Qwen2-7B|86|40|74|28|36|73|46|47|36|46.7|
> > > |LLaVA-OV|Qwen2-7B|82|51|64|34|45|71|43|45|35|48.5|
> > > |Slime|Llama3-8B|58|36|51|29|33|51|27|41|34|37.1|
> > > |Random Token Selection|LongVA-7B|68|38|37|38|40|49|33|56|38|44.0|
> > > |**UHR-BAT**|LongVA-7B|74|43|44|45|43|59|44|68|45|**50.6**|
> > >
> > > Table2. Results on HR-Bench-8K[2].
> > >
> > > |Method|Single|Cross|Overall Avg.|
> > > |:--|--:|--:|--:|
> > > |InternVL-1.5-26B|69.3|46.5|57.9|
> > > |GPT-4o|62.0|49.0|55.5|
> > > |Qwen-VL-max|54.0|51.0|52.5|
> > > |XComposer2-4kHD-7B|55.3|47.3|51.3|
> > > |LLaVA-1.6-34B|44.5|50.3|47.4|
> > > |**UHR-BAT**|64.5|51.5|**58.0**|
> > >
> > > Table3. Results on TreeBench[3].
> > >
> > > |Method|Attributes|Material|OCR|Object Retrieval|Physical State|Comparison|Contact and Occlusion|Ordering|Perspective Transform|Spatial Containment|Overall Avg.|
> > > |:--|:-|:-|:-|:-|:-|:-|:-|:-|:-|:-|:-|
> > > |InternVL3-78B|62.1|61.5|52.9|68.8|52.2|45.5|61.0|33.3|16.5|86.2|46.4|
> > > |Qwen2.5-VL-72B|65.5|69.2|48.5|56.3|56.5|38.6|51.2|33.3|11.8|72.4|42.2|
> > > |Gemini-2.5-Flash-0520|48.3|53.9|75.0|68.8|69.6|43.2|56.1|19.3|15.3|72.4|45.9|
> > > |GPT-4o-1120|51.7|61.5|69.1|43.8|65.2|43.2|48.8|38.6|18.8|72.4|46.9|
> > > |LLaVA-OneVision-72B|62.1|53.8|36.8|62.3|65.2|47.7|53.7|28.1|12.9|65.5|40.5|
> > > |DeepEyes-7B|62.1|53.8|51.5|68.8|65.2|47.7|36.6|24.6|11.8|51.7|37.5|
> > > |Pixel-Reasoner-7B|58.6|61.5|48.5|50.0|65.2|40.9|39.0|31.6|14.1|44.8|39.0|
> > > |**UHR-BAT**|51.7|53.8|47.1|50.0|**73.9**|**59.1**|**58.5**|33.3|**30.6**|58.6|**47.2**|
> > >
> > > ***
> > > We sincerely thank you for the constructive assessment. Given that most technical concerns have been addressed, **we hope the newly added transfer study and the clarified scope of the paper help demonstrate that the proposed method is not limited to the original remote-sensing setting, but addresses a broader high-resolution token-budget problem of interest to the ML community.** We will include these results in the revision and clarify that external baselines are taken from the corresponding benchmark papers or official leaderboards. **Because the Overall Recommendation score is very important to the final outcome of our submission, we would sincerely appreciate it if you could kindly reconsider your Overall Recommendation in light of the additional evidence and clarifications above.**
> > >
> > > **reference**
> > >
> > > [1] Zhang, Yi-Fan, et al. "MME-RealWorld: Could Your Multimodal LLM Challenge High-Resolution Real-World Scenarios that are Difficult for Humans?" ICLR 2025
> > >
> > > [2] Wang, Wenbin, et al. "Divide, Conquer and Combine: A Training-Free Framework for High-Resolution Image Perception in Multimodal Large Language Models" AAAI 2025
> > >
> > > [3] Haochen Wang, et al. "Traceable Evidence Enhanced Visual Grounded Reasoning: Evaluation and Methodology" ICLR 2026

---

### Official Review · Reviewer_vuxX · 2026-03-08

**Soundness:** 2
**Presentation:** 3
**Significance:** 1
**Originality:** 1
**Overall Recommendation:** 2
**Confidence:** 4

**Summary:**

This paper proposes a method to improve how vision–language models handle ultra-high-resolution remote sensing images. The approach encodes images at multiple scales using a frozen Vision Transformer (ViT) and identifies important regions through region partitioning methods such as SAM segmentation or feature-based clustering. Important regions are preserved while less informative tokens are merged via average pooling to produce a compact representation.

The resulting region tokens form a coarse visual context that is passed to a large language model (LLM) for prediction. By prioritizing salient regions and compressing redundant tokens, the method aims to maintain key semantic information while reducing the number of visual tokens and computational cost without retraining the vision backbone.

**Compliance With Llm Reviewing Policy:**

Affirmed.

**Key Questions For Authors:**

### Questions
 1. Does the proposed method reuse an existing pretrained checkpoint, or does the model need to be retrained on the ultra-high-resolution benchmark datasets?

2. How does the proposed method compare with baseline models in terms of efficiency (e.g., FLOPs, latency, or memory usage), and can the authors provide quantitative measurements to better illustrate the efficiency–performance trade-off?

3. Could the authors provide a more detailed ablation analysis to quantify the contribution of each component in the framework, including query-aware scoring, cross-scale alignment, region partitioning, and the preserve-and-merge strategy?

4. I have a significant concern regarding where the region clustering operation is performed. In the current design, it is applied immediately after the ViT module. However, the ViT itself is also a compute-bound stage (in addition to the LLM prefill stage). Why is token compression not applied earlier—either before or within the ViT encoder—to reduce the computational burden of the vision backbone?

5. Also see weaknesses.

**Limitations:**

The paper emphasizes empirical improvements but provides relatively little discussion about scenarios where the proposed token compression strategy might fail or perform poorly.

**Strengths And Weaknesses:**

### Strengths
- Overall the paper is clearly written, and the problem setting and motivation are well articulated, making the approach easy to follow.
- The paper conducts extensive experiments and evaluates the method against a large number of baselines, including both open-source and closed-source models.

### Weaknesses
- **Dependency on external partitioning:** The region partitioning step may rely on SAM-based segmentation or K-Means clustering, which introduces additional system complexity and large potential runtime overhead. The sensitivity to different partitioning strategies is also not thoroughly analyzed.

- **Limited ablation:** The method includes several modules (query-aware scoring, cross-scale alignment, region partitioning, and preserve-and-merge), but the ablation study mainly varies hyperparameters such as cluster number and token budget, leaving the impact of individual components unclear.


- **Limited comparison with related token compression methods:** The paper compares against several VLM baselines but lacks evaluation against recent token pruning or token merging methods such as  ToMe (Bolya et al., 2023), PiToMe (Tran et al., 2024), ToFu (Kim et al., 2023), which are closely related to the proposed compression strategy.

- **Limited efficiency analysis:** Although the paper targets efficiency, it does not report FLOPs, latency, or memory usage for baseline models, making the accuracy–efficiency trade-off difficult to assess.

# Reference

ToMe- Bolya, D., Fu, C.-Y., Dai, X., Zhang, P., & Feichtenhofer, C. (2023). Token Merging: Your ViT but Faster.

PiToMe- Tran, H.-C., et al. (2024). Accelerate Transformer with Spectrum Preserving Token Merging

ToFu- Kim, S., et al. (2023). ToFu: Token Fusion for Efficient Vision Transformers.

---

> ### Author Rebuttal · Authors · 2026-03-31
>
> We thank **reviewer vuxX** for the positive feedback on our writing, problem formulation, motivation, and extensive experiments. We appreciate the suggestions and will further discuss the limitations of our method in the revision.
> ***
> Q1: Sensitivity to different partitioning strategies is not thoroughly analyzed.
>
> A1: We conducted experiments evaluating both different partitioning methods (SAM, BIRCH, K-means) and varying numbers of clusters ($K=20,50,100,200$). As shown in Tables 1 and 2, our method is robust to different partitioning strategies.
>
> Table1. Robustness Across Partitioning Methods ($N$ means retained tokens.)
> |Method|$N=4k$|$N=6k$|$N=8k$|
> |:-|:-|:-|:-|
> |K-means|43.3|43.4|43.7|
> |SAM|43.3|43.5|43.6|
> |BIRCH|43.1|43.3|43.3|
> |Max $\Delta$|0.2|0.2|0.4|
>
> Table2. Robustness Across Cluster Counts ($K$)
> |Metric|$K=20$|$K=50$|$K=100$|$K=200$|Max $\Delta$|
> |:-|:-|:-|:-|:-|:-|
> |Perception|41.1|40.9|41.6|41.3|0.7|
> |Reasoning|58.1|58.3|57.4|57.8|0.9|
> |Overall|43.7|43.5|**44.0**|43.8|**0.5**|
> ***
> Q2: SAM-based segmentation or K-Means clustering may introduce additional system complexity and large runtime overhead.
>
> A2: We respectfully clarify that SAM is used strictly offline during training to provide high-quality region priors; **only lightweight K-Means clustering is used during online inference**. So, SAM does not impact the actual deployment efficiency.
> Experiments show that the extra cost of K-Means during inference is **negligible relative to the overall system** (Due to the space, detailed reuslts in **reviewer fqwg (Q8-A8-Table6)**).
> ***
> Q3: Could the authors provide a more detailed ablation of each component?
>
> A3: Following your suggestion, we conducted comprehensive component-wise ablations to examine the contribution of each module. (Due to the space, detailed reuslts in **reviewer gCRt (Q2-A2)**). Removing any component **consistently degrades performance**.
> ***
> Q4: The paper lacks evaluation against recent methods such as ToMe, PiToMe, ToFu.
>
> A4: Following your constructive feedback, we conducted extensive comparisons with these methods. Our method achieves the best performance on all evaluation dimensions. And we will include it in the experiment section of the revision.
>
> Table3. Comparison with other Token Reduction Methods
> |Category|Method|Perception|Reasoning|Acc.|
> |:-|:-|:-|:-|:-|
> |In-ViT|ToMe|37.7|53.4|40.0(-4.0)|
> |In-ViT|PiToMe|37.7|53.9|40.1(-3.9)|
> |In-ViT|ToFu|38.0|50.2|39.8(-4.2)|
> |Plug-in|AIM|39.2|57.4|42.0(-2.0)|
> |RS-Specific|RFM+DIP|31.7|54.3|35.0(-9.0)|
> |Ours|UHR-BAT|41.5|57.8|**44.0**|
> ***
> Q5: How does the method compare with baseline in terms of efficiency? Can the authors provide quantitative measurements?
>
> A5: We added quantitative efficiency analysis. Our method turns an otherwise undeployable baseline into an efficient model, while maintaining **a strong accuracy-efficiency trade-off compared to naive ablation variants** (Table4).
>
> Table4. Quantitative measurements
> |Variant|Acc.|TFLOPs|Memory(GB)|Latency(s)|
> |:-|:-:|:-|:-|:-|
> |Baseline (Full Tokens)|-|8650*|128.9*|180.0*|
> |Ours|**44.0**|272.7 (↓96.8%)|24.6 (↓80.9%)|9.0 (↓95.0%)|
> |w/o Cross-scale alignment|42.9|268.5|23.5|8.0|
> |w/o Region partitioning|42.7|268.5|23.5|8.7|
> |w/o Preserve-and-merge|42.8|266.4|23.5|7.7|
> |w/o Query-aware scoring|42.2|125.0|22.1|6.4|
> ***
> Q6: Does the method reuse a pretrained checkpoint or require retraining?
>
> A6: We reuse a pretrained checkpoint. As detailed in **Appendix B** (More Details of Our Method, Line 1), UHR-BAT is initialized from LLaVA-Next, with CLIP-ViT-L/14-336 as the vision encoder and LongVA-7B as the language backbone. It is then fine-tuned on only 10K image-text pairs sampled from SuperRS-VQA and HighRs-VQA.
> ***
> Q7: Why is token compression not applied earlier—either before or within the ViT encoder?
>
> A7: There are two main reasons: 1. Early compression is performed **before explicit query-conditioned relevance is available**, which may lead to neglect of the text query. 2. UHR images contain massive backgrounds and tiny sparse targets, so compression before the ViT or merging in early ViT layers can easily absorb small targets into dominant backgrounds (e.g., a small boat into the ocean).
>
> To quantify the preservation of query-critical visual features, we define the **Attention Retention Rate (ARR)** as the ratio of the retained tokens' text-to-vision attention scores to the global attention sum:
> $ARR = \frac{\sum\_{i\in\mathcal{S}\_{kept}}a\_i}{\sum\_{i=1}^Na_i}\times 100$%.
> The results show that inside-ViT compression yields **much lower ARR** than our post-ViT design, consistent with substantially **lower accuracy**.
>
> Table5. Quantitative Comparison of Token Compression Strategies
>
> |Method|Stage|ARR (%)|Overall Acc.|
> |:-|:-:|:-:|:-:|
> |ToMe|Inside ViT|5.12|40.0 (-4.0)|
> |PiToMe|Inside ViT|6.09|40.1 (-3.9)|
> |ToFu|Inside ViT|5.48|39.8 (-4.2)|
> |Ours|After ViT|12.85|**44.0**|
>
> Note: Baseline values are estimated since it does not fit on an 80GB GPU; others are measured on one A100.

---

> > ### Author Rebuttal · Reviewer_vuxX · 2026-04-04
> >
> > Thank you for the detailed rebuttal and the provided efficiency gain breakdown. While the added data is helpful, I still have significant concerns regarding the justifications for the proposed architecture and the fairness of the baseline comparisons.
> >
> > 1. Fairness of Comparison (In-ViT vs. Post-ViT).
> >
> > The comparison with recent token compression methods (ToMe, PiToMe, ToFu) remains problematic. The authors argue these are "In-ViT" methods, but do not sufficiently explain why they could not be adapted as "Post-ViT" baselines to ensure a head-to-head comparison with the proposed method, because these are just plug-and-play algorithms. It is difficult to discern whether the performance gains are due to your specific algorithm or simply the advantage of performing compression after the vision backbone.
> >
> > 2. Query-Conditioned Relevance Timing.
> >
> > I find the claim that early compression "may lead to neglect of the text query" unpersuasive. A query-conditioned relevance is essentially a prompt/embedding that can be introduced at any stage. Many existing methods [1, 2] successfully use text queries to filter or select image patches within various layers of the ViT.
> >
> > 3. Target Preservation and Feature Abstraction.
> >
> > The authors argue that early-layer compression absorbs small targets into dominant backgrounds. While early layers are indeed sensitive, visual tokens become increasingly abstract and high-level in deeper layers. This is precisely why most training-free compression methods employ a gradual reduction strategy across layers rather than aggressive pruning/merging at the very beginning. The assertion that early-stage compression is inherently unsuitable for Ultra-High-Resolution (UHR) imagery lacks strong empirical support in the rebuttal, especially when compared to a gradual compression approach.
> >
> > References
> >
> > - [1] CrossGET: Cross-Guided Ensemble of Tokens for Accelerating Vision-Language Transformers
> > - [2] TRIPS: Efficient Vision-and-Language Pre-training with Text-Relevant Image Patch Selection
> >
> >
> > Given these unresolved points, I will keep my score as is.

---

> > > ### Author Response · Authors · 2026-04-07
> > >
> > > We thank the **reviewer vuxX** for the thoughtful feedback. We agree that our previous rebuttal did not sufficiently disentangle three factors: compression stage, query-conditioning timing, and gradual reduction. We therefore added controlled experiments under the same **UHR** setting and matched final token budget.
> > > ***
> > > Q1: Is the gain simply from using Post-ViT compression rather than from the proposed algorithm itself?
> > >
> > > A1: To isolate this factor, we implemented adapted **Post-ViT** versions of **ToMe / PiToMe / ToFu** by taking their token merging or fusion operators and applying them **after the final ViT block**, under the **same final token budget** as our method. These variants serve as stage-controlled baselines.
> > >
> > > The results show that moving the same operator from **In-ViT** to **Post-ViT** indeed helps in UHR settings, so the reviewer’s concern about compression timing is valid. However, our method still outperforms the best adapted Post-ViT baseline by **+3.0 overall**. Therefore, the gain cannot be explained solely by the structural placement of compression after the backbone.
> > >
> > > |Method|Perception|Reasoning|Overall|
> > > |-|-:|-:|-:|
> > > |In-ViT ToMe|37.7|53.4|40.0|
> > > |In-ViT PiToMe|37.7|53.9|40.1|
> > > |In-ViT ToFu|38.0|50.2|39.8|
> > > |Post-ViT ToMe|38.0|56.7|40.8|
> > > |Post-ViT PiToMe|38.1|57.6|41.0|
> > > |Post-ViT ToFu|37.9|57.6|40.9|
> > > |**Ours**|41.5|57.8|**44.0**|
> > > ***
> > > Q2: If query-conditioned relevance can be introduced early, why not use early-/mid-fusion selection instead?
> > >
> > > A2: We agree that query-conditioned relevance can be injected early. The key issue is not whether the query can be introduced early, but whether **early irreversible compression can reliably preserve tiny, answer-critical evidence**. We tested query-guided baselines and a timing ablation. For TRIPS, we reproduced the method described in the paper, **unfroze the ViT backbone as required by its design**, and **re-finetuned on the same dataset using six A100 GPUs**.
> > >
> > > |Method|Perception|Reasoning|Overall|
> > > |-|-:|-:|-:|
> > > |TRIPS|38.8|56.5|41.5|
> > > |CrossGET + our checkpoint|38.3|56.7|41.1|
> > > |**Ours**|**41.5**|**57.8**|**44.0**|
> > >
> > > We further conducted a controlled ablation by **applying the full UHR-BAT pipeline inside the ViT backbone**. Specifically, we preserved the same **query conditioning, token-importance scoring, compression policy**, and **final 11k token budget**, and changed only the stage at which compression was performed. Thus, this ablation isolates the effect of **compression timing**, rather than differences in query formulation or scoring design.
> > > To quantify target preservation, we report two additional metrics using the dataset-provided target region h_box as the answer-critical region: **Target Recall**, the fraction of examples with at least one retained token inside the region, and **Coverage**, the fraction of the region covered by retained tokens.
> > > These results consistently show that delaying irreversible compression improves both accuracy and target preservation.
> > >
> > > |Stage|Overall|Target Recall|Coverage|
> > > |-|-:|-:|-:|
> > > |Block 2|23.6|0.073|0.005|
> > > |Block 6|30.1|0.238|0.049|
> > > |Block 12|36.2|0.283|0.060|
> > > |Block 20|41.8|0.567|0.372|
> > > |Final Block|42.3|0.954|0.496|
> > > |**Ours**|**44.0**|**0.985**|**0.795**|
> > > ***
> > > Q3: Early layers are sensitive, but why is gradual compression still not enough?
> > >
> > > A3: We agree that gradual reduction is more reasonable than aggressive one-shot early compression, and we explicitly tested it. We performed staged compression at **Blocks 2/6/12/20** under the same final **11k-token budget**. The results show that while **gradual compression is better than aggressive early compression**, it still remains clearly below **late/post-ViT compression** in both accuracy and target preservation.
> > >
> > > |Method|Perception|Reasoning|Overall|Target Recall|Coverage|
> > > |-|-:|-:|-:|-:|-:|
> > > |Gradual (2/6/12/20)|36.5|54.3|39.2|0.304|0.138|
> > > |Final Block|39.6|57.6|42.3|0.954|0.496|
> > > |**Ours**|41.5|57.8|**44.0**|**0.985**|**0.795**|
> > > ***
> > > Therefore, we do **not** argue that early compression is universally unsuitable. Rather, our evidence supports the narrower conclusion that, under **frozen ViT + UHR imagery**, where each input starts from **over 130k visual tokens** and must be compressed to a **strict final budget**, delaying irreversible compression is empirically more reliable for preserving tiny, answer-critical targets.
> > > ***
> > > We also appreciate the reviewer for highlighting these highly relevant references. In the revised manuscript, we will explicitly cite and discuss **ToMe, PiToMe, ToFu, CrossGET**[1] and **TRIPS**[2] as representative token compression approaches.
> > >
> > > We are sincerely grateful for your careful reassessment. We hope that the newly added controlled experiments and the narrowed scope of our claims make our contribution clearer and better supported. **If you feel that these additional results and clarifications have addressed your remaining concerns, we would sincerely appreciate your reconsideration of the current overall recommendation.**

---

### Official Review · Reviewer_gCRt · 2026-03-12

**Soundness:** 3
**Presentation:** 3
**Significance:** 3
**Originality:** 3
**Overall Recommendation:** 5
**Confidence:** 3

**Summary:**

The paper presents a method for ultra-high-resolution remote sensing that processes the input image at multiple scales and assigns a limited token budget for each scale and each detected region inside of it, ensuring that representative regions of the image have at least one token. The method achieves a higher or a comparable performance to other existing methods on different tested benchmarks.

**Compliance With Llm Reviewing Policy:**

Affirmed.

**Final Justification:**

The authors have performed additional experiments that address my questions and concerns and reacted accordingly to the mistakes found in the text.

**Key Questions For Authors:**

- What is the reason to not include token importance when merging tokens in Equation 10?

**Limitations:**

- Ablation study is limited to the choice of parameters for the proposed modules and does not cover the replacement of proposed modules.

**Strengths And Weaknesses:**

Strengths
- The paper is well-structured and written;
- Extensive experiments show the improvements in average model performance compared to open-source SOTA.

Weaknesses
- The inference time analysis lacks a comparison with the existing methods. Also, the choice for minimum compression is not explained. If it is limited by available memory, it should be explicitly shown. Otherwise, the evaluation should also be done with no compression present. VRAM use with different compression levels should also be shown in the experiments since this is one of the main limitations when applying ML models to edge devices.
- Scales are referenced before Section 3.3 but are explicitly defined only in this section, which reduces text clarity.
- $\tau_s(u,v)$ is not defined.
- Figure 2 is not referenced.
- Tables 2 and 3. The best and second-best results should be highlighted.

---

> ### Author Rebuttal · Authors · 2026-03-31
>
> We thank **reviewer gCRt** for the positive feedback on the paper’s structure, clarity, extensive experiments, and the demonstration of state-of-the-art performance improvements.
> Below, we respond point by point to the concerns.
> ***
> Q1: Why not use token importance in Eq. 10 merging?
>
> A1: We thank the reviewer for pointing this out, which inspired this additional analysis. We compared three settings: no merging, importance-based merging, and average merging. The results indicate that importance merging may introduce **prohibitive computational overhead without yielding significant performance improvements** within our specific framework.
>
> Table1. Comparison of different token merging strategies on model performance.
> |Method|Perception Accuracy|Reasoning Accuracy|Overall Accuracy|
> |:-|:-|:-|:-|
> |Without merging token|40.2|57.6|42.8|
> |Importance-merging|41.4(+1.2)|57.6(+0.0)|43.8(+1.0)|
> |Avg-merging (Ours)|41.5(+1.3)|57.8(+0.2)|**44.0(+1.2)**|
>
> To deeply understand why importance weighting has such a minimal impact, we **investigated the attention score distributions**. Tokens merged by our pruning rule are already highly similar. We hypothesized that their attention scores must also be nearly identical. To quantify this, we measured the variance of attention scores:
> - **Global Variance** (across all tokens):
> $3.043 \times 10^{-9}$
> - **Mean Internal Variance within merge groups**: $1.209 \times 10^{-10}$ (~25$\times$ smaller)
> ***
> Q2: Ablation study does not cover module replacement.
>
> A2: We conducted component-wise ablations under the same experimental settings by removing each core module. As the results demonstrate, **removing any single component consistently leads to a noticeable performance drop**.
>
> Table2. Ablation Study of Core Components
> |Variants|Overall|Perception|Reasoning|
> |:-|:-:|:-:|:-:|
> |w/o Query-aware scoring|42.2(-1.8)|39.8(-1.7)| 55.9(-1.9)|
> |w/o Cross-scale alignment|42.9(-1.1)|40.4(-1.1)|57.2(-0.6)|
> |w/o Region partitioning|42.7(-1.3)|40.1(-1.4)| 57.6(-0.2)|
> |w/o Preserve-and-merge|42.8(-1.2)|40.2(-1.3)|57.6(-0.2)|
> |Full Model (Ours)|**44.0**|41.5|57.8|
> ***
> Q3：The inference time analysis lacks a comparison with the existing methods.
>
> A3: Following your valuable suggestion, we added comparisons with both the **general MLLM** and  recent **pruning methods**. The results demonstrate that our method achieves **the lowest total latency** among them.
>
> Table3. Comparison of Inference Time
> |Method|Avg. Gen.(s)|Avg. Total(s)|
> |:-|:-|:-|
> |Qwen2.5-VL-7B|15.52|21.75 (+12.71)|
> |AIM([2])|3.11|15.60 (+6.56)|
> |RFM+DIP([1])|2.98|54.54 (+45.50)|
> |**Ours**|2.96|**9.04**|
> ***
> Q4: The choice for minimum compression is not explained. VRAM use with different compression levels should also be shown.
>
> A4: We re-evaluated our method under different compression levels to figure out minimum compression. We tried to preserve as much tokens as we could. The results demonstrate that our selection strategy drastically reduce the VRAM usage and latency with only **marginal accuracy degradation**, thereby proving its high potential for edge device deployment.
>
> Table4. Performance and Resource Trade-offs across Different Compression Levels
> |Kept Tokens|Latency (s)|Accuracy|TFLOPS|Usage (GB)|
> |:-|:-|:-|:-|:-|
> |131k (No Compression)|180.0*|OOM|8650.0*|128.85*|
> |44k|24.54|44.5|816.0|84.05|
> |11k|14.48|44.0(-0.5)|358.0|29.57|
> |8k|10.38|43.7(-0.8)|315.3|26.73|
> |5k|9.04|43.5(-1.0)|272.7|24.59|
>
> *\* Analytically estimated due to OOM in the uncompressed 131K setting.*
> ***
> Q5：(1) Scales are referenced before Section 3.3 but are explicitly defined only in this section. $τ_s(u,v)$ is not defined. (2) Figure 2 is not referenced.(3) Tables 2 and 3. The best and second-best results should be highlighted.
>
> A5：Thank you for pointing these out. We apologize for our oversight. We will correct them in our paper.
> (1) Firstly, we will introduce the multi-scale setting earlier in Section 3.2 (“Multi-Scale Formulation”), stating that the input image is resized into $S$ views {$I^{(s)}$}$\_{s=1}^S$ where s = 1 is the anchor scale preserving global structure. We will also define the resize mapping $τ\_s(u,v)$ in Section 3.3 when Eq.(6) is introduced, clarifying that it maps a grid location at scale s to the corresponding continuous coordinate on the anchor-scale grid for bilinear interpolation.
> (2) We will also add a reference to Figure 2 at the beginning of the second paragraph in Section 3.1 (Overview).
> (3) Finally, we promise to highlight the best and second-best results in our Table 2 and 3 to make the comparison more clear.
>
> **Note: For a fair comparison, the runtime, memory usage, and FLOPs of all the aforementioned models are measured on an NVIDIA A100 GPU.**
> #### References
> [1] Luo, Junwei, et al. "When large vision-language model meets large remote sensing imagery: Coarse-to-fine text-guided token pruning." ICCV 2025.
>
> [2] Zhong, Yiwu, et al. "Aim: Adaptive inference of multi-modal llms via token merging and pruning." ICCV 2025.

---

> > ### Author Rebuttal · Reviewer_gCRt · 2026-04-04
> >
> > Thank you for the additional analysis and experiments.

---

> > > ### Author Response · Authors · 2026-04-07
> > >
> > > Thank you very much for reading our rebuttal and we are glad that your concerns have been resolved. We sincerely value your suggestions regarding token merging methods, ablation studies, and comparative experiments. We will be sure to include the above points in our revision, and we will certainly refine the phrasing in our paper to improve its clarity. Thank you once again for your time and advice.

---

### Official Review · Reviewer_fqwg · 2026-03-15

**Soundness:** 3
**Presentation:** 3
**Significance:** 3
**Originality:** 2
**Overall Recommendation:** 4
**Confidence:** 4

**Summary:**

This paper addresses the challenge of applying multimodal large language models (MLLMs) to ultra-high-resolution (UHR) remote sensing imagery. The authors propose UHR-BAT, a token compression framework built on two principles: query-guided compression (retaining tokens relevant to the input question) and region-faithful compression (preserving coverage of semantically coherent regions, including sparse small targets). In practice, the method utilizes text-to-vision cross-attention scores to estimate token importance, constructs multi-scale image representations with scale-specific positional embeddings and cross-scale importance alignment, and implements a Region-wise Preserve-and-Merge (RPM) strategy that selectively retains high-importance tokens within semantic regions while compressing redundant background tokens via mean pooling. The approach is evaluated on XLRS-Bench, RSHR-Bench, and MMERealWorld-RS datasets.

**Compliance With Llm Reviewing Policy:**

Affirmed.

**Key Questions For Authors:**

- Ablation of RPM vs. multi-scale encoding. Tables 6 and the ablation in Figure 3 vary the token budget and cluster count, but there is no experiment that uses the multi-scale encoding without RPM (e.g., replacing it with standard global top-K pruning). How much of the performance improvement over baselines like GeoLLaVA-8K is attributable specifically to RPM, and how much to the multi-scale representation?

- Faithfulness of attention-based and qualitative justifications. The pipeline exposes some of its intermediate reasoning through the attention maps in Figure 8, but a correct answer is not proof that the model attends to the right regions for the right reasons. Attention maps are known to be imperfect proxies for causal importance. Furthermore, the qualitative examples in Figure 12 appear to be curated, making it difficult to assess their representativeness. Could the authors provide a more systematic validation, for instance, by measuring accuracy degradation when the top-attended tokens are masked out, or by reporting the selection criteria for the qualitative examples?

- Random token sampling baseline. It seems that no experiment compares the proposed importance-guided selection against random token sampling at the same budget. Without this baseline, it is difficult to isolate how much of the performance gain is attributable to the attention-guided and region-aware selection strategy, rather than simply to the multi-scale encoding. Could the authors provide this comparison?

- Computational overhead of region partitioning. The paper notes that SAM is too slow for practical use and prefers clustering. Table 4 reports total and generation latency, but does not include the time required for K-means clustering over the token grid. Could you provide the end-to-end wall-clock time including preprocessing for both partitioning methods?

- Budget allocation across scales. The per-scale budget {B_s} is described as assigning larger budgets to higher-resolution scales, but the exact allocation scheme is described in the appendix with specific values for each benchmark separately. Is there a principled rule for setting these values, or are they tuned per benchmark? How sensitive is performance to this choice?

**Limitations:**

The authors include an Impact Statement discussing resource efficiency and edge deployment, but do not provide a dedicated limitations section.

**Strengths And Weaknesses:**

Soundness:

On the positive side, the technical design is coherent and internally consistent. The ablation studies are reasonably thorough, covering most of the critical features. The results generally support the design choices. The efficiency analysis in Table 4 concretely demonstrates the trade-off between compression ratio, FLOPs and latency.

On the negative side, the evaluation relies exclusively on multiple-choice VQA benchmarks. No open-ended generation or grounding experiments are presented, which limits the generalizability of the claims. More importantly, the paper lacks a head-to-head controlled ablation that isolates the contribution of RPM from the multi-scale encoding: it is unclear how much gain comes from the region-aware merging versus simply using multiple resolutions.

Another issue is that the model is fine-tuned using SuperRS-VQA and HighRs-VQA samples, and then evaluated using XLRS-Bench and RSHR-Bench. The degree of distributional overlap between the fine-tuning data and the benchmarks is not reported. This makes it difficult to determine how much of the improvement is due to the compression mechanism versus domain adaptation.


Presentation:

The paper is generally well-written. Figure 2 provides a clear overview of the pipeline.

The appendix is well detailed and covers pseudocode, benchmark descriptions, and implementation specifics. However it is substantially longer than the main paper; some of the pseudocode and derivations could be condensed. The paper compares against GeoLLaVA-8K (a direct competitor in the UHR remote sensing space) but the comparison at matched token budgets is not shown explicitly in the main text, which would strengthen the efficiency claim.



Originality:

The individual components (cross-attention importance scoring, multi-scale encoding, clustering-based region partition, token merging) each have precedents in the literature (e.g., GeoLLaVA-8K, ZoomEarth, LLaVA-UHD). The contribution lies in their integration under a unified hard-budget framework with the specific RPM mechanism and cross-scale importance alignment.

The related work section, while covering relevant prior work, reads like a catalogue of citations rather than clearly articulating the precise technical gap this work addresses relative to the most closely related methods (GeoLLaVA-8K, ZoomEarth, ZoomSearch). A direct, side-by-side comparison of the core mechanisms would strengthen positioning.



Significance:

The problem of efficient visual reasoning over very large satellite and aerial images is practically relevant for geospatial applications. However, its scope is specialized to this field and would likely require re-tuning for other domains.


Overall Assessment:

UHR-BAT is a solid, well-motivated engineering paper that addresses a genuine and practically important problem. The method is technically coherent, the experiments are broad, and the results are competitive. However, the novelty is incremental: the core contribution is a careful integration of existing ideas rather than a fundamental methodological advance, and the scope specialized to remote sensing.

---

> ### Author Rebuttal · Authors · 2026-03-31
>
> We sincerely thank **reviewer fqwg** for the positive feedback on our technical design, ablations, efficiency analysis, writing, Figure 2, and appendix. Below, we respond point by point to the concerns.
> ***
> Q1: Controlled ablation is missing.
>
> A1: We conducted **component-wise ablations** under the same setting by removing each core module. The results show that **removing any single component consistently leads to a noticeable performance drop**:
>
> Table1. Ablation Study of Core Components
> |Variants|Overall|Perception|Reasoning|
> |:-|:-|:-|:-|
> |w/o Query-aware scoring|42.2(-1.8)|39.8(-1.7)|55.9(-1.9)|
> |w/o Cross-scale alignment|42.9(-1.1)|40.4 (-1.1)|57.2(-0.6)|
> |w/o Region partitioning|42.7(-1.3)|40.1(-1.4)|57.6(-0.2)|
> |w/o Preserve-and-merge|42.8(-1.2)|40.2(-1.3)|57.6(-0.2)|
> |Ours|**44.0**|41.5|57.8|
> ***
> Q2: Need a more systematic validation of the proposed importance-guided selection, including masking top-attended tokens and comparing against random sampling.
>
> A2: Following your constructive feedback, we conducted the requested validations, including **random-baseline comparisons** and a **mask-out experiment** that removes the 80, 320, 600, and 2000 tokens identified by UHR-BAT as most important. We compared four selection strategies under the same per-scale token budget. Table 2 shows that random selection substantially underperforms our method, and masking out top-attended tokens performs even worse than random regional selection.
>
> Table2. Ablation on Token Selection Strategies
> |Strategy|Perception Acc.|Reasoning Acc.|Overall|
> |:-|:-|:-|:-|
> |Random Token Selection|37.6|54.1|40.1|
> |w/o Query-aware scoring|39.8|55.9|42.2|
> |Maskout|39.6|55.8|42.0|
> |Ours|41.5|57.8|**44.0**|
> ***
> Q3: No grounding or open-ended generation results.
>
> A3: We added evaluations on these tasks. In XLRS-Bench, Object Motion State (OMS), Object Classification (OCC), and Object Color Recognition (OCR) test visual grounding. Also, we evaluated on the LRS-VQA.
>
> Table3. Evaluation of visual grounding and open-ended tasks.
> |Model|OMS|OCC|OCR|LRS-VQA|Avg. Acc.|
> |:-|:-:|:-:|:-:|:-:|:-:|
> |LLaVA-Next|28.8|32.8|66.7|55.0|45.8|
> |GeoLLaVA-8K|41.6|31.6|65.0|56.2|48.6|
> |Ours|43.5|33.8|65.0|65.6|**52.0**|
> ***
> Q4: Rule and sensitivity of per-scale budget $B\_s$.
>
> A4: Thank you for pointing this out. The budget is allocated in proportion to the number of visual tokens at each resolution scale. **Table 6 in the paper** shows that performance is robust across different scale-specific budgets, and proportional allocation is the most stable and effective choice.
> ***
> Q5: Overlap between fine-tuning data and benchmark.
>
> A5: Thank you for raising this important concern.
> The overlap is only at the **source-dataset level**; we did **not** fine-tune on XLRS-Bench samples. Specifically, the two splits share three source datasets (DOTA, HRSCD, and MiniFrance).
>
> Source-level overlap summary
> |Split|Shared Sources|Unique Sources|
> |:-|:-|:-|
> |Fine-tuning|DOTA + HRSCD + MiniFrance: 9117 (91.2%)|ft3_synthetic: 883 (8.8%)|
> |XLRS-Bench|DOTA + HRSCD + MiniFrance: 2957 (96.0%)|ITCVD + Potsdam + Toronto: 123 (4.0%)|
>
> To isolate domain adaptation from our compression method, we include a **controlled baseline** trained with the same protocol but without our method (Table 4). Our method yields about 2× the gain of dataset adaptation alone, so the improvement cannot be explained by domain adaptation only.
>
> Table4. Controlled analysis of domain adaptation vs. our method
> |Model|Perception Acc.|Reasoning Acc.|Overall|
> |:-|:-|:-|:-|
> |LLaVA-Next (Baseline)|37.2|54.5|39.8|
> |+ RS Fine-tuning Data|38.3|57.6|41.2(+1.4)|
> |+ RS Fine-tuning Data + Our Method|41.5|57.8|**44.0(+4.2)**|
> ***
> Q7: No explicit matched-budget comparison with GeoLLaVA-8K.
>
> A7: Following the suggestion, we added matched-budget comparisons with GeoLLaVA-8K from 4K to 8K tokens. Across all budgets, our method consistently outperforms GeoLLaVA-8K while remaining stable.
>
> Table5. Performance comparison at matched token budgets
> |Token Budget|4K|6K|8K|
> |:-|:-|:-|:-|
> |GeoLLaVA-8K|42.2|42.3|42.5|
> |Ours|43.3(+1.1)|43.5(+1.2)|43.6(+1.1)|
> ***
> Q8: Provide the end-to-end wall-clock time, including preprocessing, for both partitioning methods.
>
> A8: Thank you for this important question. The inference time reported in the paper already **reflects the true end-to-end wall-clock time**, including K-means clustering, which is executed online during inference. By contrast, SAM is **used only offline** during training to produce region priors. To make this clearer, we additionally report the K-means overhead below, which is **negligible**.
>
> Table6. Efficiency Analysis of SAM and K-Means
> |Method|Setting|Phase|Time(s)|Memory(GB)|TFLOPS|
> |-|-|-:|-:|-:|-:|
> |SAM|Offline|Training|1.15|22.07|8.86|
> |K-Means|Online|Inference|0.36(~3.9%)|0.93(~3.7%)|5.14(~1.8%)|
> |Overall|Online|Inference|9.04|24.59|272.7|
>
> For a fair comparison, the runtime, memory usage, and FLOPs of all the aforementioned models are measured on an NVIDIA A100 (80GB) GPU.

---

### Decision · Program_Chairs · 2026-04-30

**Decision:**

Accept (regular)

**Comment:**

This paper proposes UHR-BAT, a token compression framework for applying multimodal LLMs to ultra-high-resolution remote sensing imagery, combining query-guided compression with region-faithful preservation. The method is coherent and well-engineered, with competitive results and reasonably thorough efficiency analysis and ablations.

However, several concerns are raised. The novelty is somewhat incremental, largely integrating existing ideas (e.g., token importance, multi-scale encoding, merging). The evaluation is incomplete, with missing comparisons to closely related methods, limited component isolation, and insufficient efficiency baselines. Questions about fairness and design choices (e.g., post-ViT compression) also remain, and the scope may be relatively narrow.

The rebuttal provides partial clarifications and some additional evidence, but does not fully resolve issues around comparison completeness and justification. Reviewer opinions are split, though generally positive.

Overall, despite limitations in novelty and evaluation rigor, the method is practically effective with consistent empirical gains. I believe it meets the bar as a solid engineering contribution and lean toward weak accept.